# Effects of Waterlogging on Soybean Rhizosphere Bacterial Community Using V4, LoopSeq, and PacBio 16S rRNA Sequence

Taobing Yu,[a,b] Lang Cheng,[a,b] Qi Liu,[a,b] Shasha Wang,[a,b] Yuan Zhou,[c] Hongbin Zhong,[c] Meifang Tang,[c] Hai Nian,[a,b] Tengxiang Lian[a,b]

[a]The State Key Laboratory for Conservation and Utilization of Subtropical Agro-bioresources, South China Agricultural University, Guangzhou, People's Republic of China
[b]The Key Laboratory of Plant Molecular Breeding of Guangdong Province, College of Agriculture, South China Agricultural University, Guangzhou, People's Republic of China
[c]BGI Genomics, BGI-Shenzhen, Shenzhen, China

**ABSTRACT** Waterlogging causes a significant reduction in soil oxygen levels, which in turn negatively affects soil nutrient use efficiency and crop yields. Rhizosphere microbes can help plants to better use nutrients and thus better adapt to this stress, while it is not clear how the plant-associated microbes respond to waterlogging stress. There are also few reports on whether this response is influenced by different sequencing methods and by different soils. In this study, using partial 16S rRNA sequencing targeting the V4 region and two full-length 16S rRNA sequencing approaches targeting the V1 to V9 regions, the effects of waterlogging on soybean rhizosphere bacterial structure in two types of soil were examined. Our results showed that, compared with the partial 16S sequencing, full-length sequencing, both LoopSeq and Pacific Bioscience (PacBio) 16S sequencing, had a higher resolution. On both types of soil, all the sequencing methods showed that waterlogging significantly affected the bacterial community structure of the soybean rhizosphere and increased the relative abundance of *Geobacter*. Furthermore, modular analysis of the cooccurrence network showed that waterlogging increased the relative abundance of some microorganisms related to nitrogen cycling when using V4 sequencing and increased the microorganisms related to phosphorus cycling when using LoopSeq and PacBio 16S sequencing methods. Core microorganism analysis further revealed that the enriched members of different species might play a central role in maintaining the stability of bacterial community structure and ecological functions. Together, our study explored the role of microorganisms enriched at the rhizosphere under waterlogging in assisting soybeans to resist stress. Furthermore, compared to partial and PacBio 16S sequencing, LoopSeq offers improved accuracy and reduced sequencing prices, respectively, and enables accurate species-level and strain identification from complex environmental microbiome samples.

**IMPORTANCE** Soybeans are important oil-bearing crops, and waterlogging has caused substantial decreases in soybean production all over the world. The microbes associated with the host have shown the ability to promote plant growth, nutrient absorption, and abiotic resistance. High-throughput sequencing of partial 16S rRNA is the most commonly used method to analyze the microbial community. However, partial sequencing cannot provide correct classification information below the genus level, which greatly limits our research on microbial ecology. In this study, the effects of waterlogging on soybean rhizosphere microbial structure in two soil types were explored using partial 16S rRNA and full-length 16S gene sequencing by LoopSeq and Pacific Bioscience (PacBio). The results showed that full-length sequencing had higher classification resolution than partial sequencing. Three sequencing methods all indicated that rhizosphere bacterial community structure was significantly impacted by waterlogging, and the relative abundance of *Geobacter* was increased in the rhizosphere in both soil types after suffering waterlogging. Moreover, the core microorganisms obtained by different sequencing methods all contain species related

**Ad Hoc Peer Reviewer** Glade Dlott

Address correspondence to Tengxiang Lian, liantx@scau.edu.cn, or Hai Nian, hnian@scau.edu.cn.

The authors declare no conflict of interest.

to nitrogen cycling. Together, our study not only explored the role of microorganisms enriched at the rhizosphere level under waterlogging in assisting soybean to resist stress but also showed that LoopSeq sequencing is a less expensive and more convenient method for full-length sequencing by comparing different sequencing methods.

**KEYWORDS** soybean, waterlogging, rhizosphere bacterial community, V4 16S rRNA sequence, LoopSeq, PacBio

Significant decline in crop yields has been witnessed globally as a result of extreme environments, such as waterlogging, drought, and extreme temperatures, in the past few decades (1, 2). Waterlogging has been considered to be one of the abiotic stresses that can adversely affect the physiological growth of plants (3–6). In particular, soil physicochemical properties (e.g., porosity, structure, and pH) will be the worst affected, where a sharp decrease in soil oxygen concentrations will have a negative impact on microbial diversity and community activity (7, 8). The loss of nitrogen (N) in waterlogged soil combined with other harmful effects, such as root hypoxia, are expected to lead to the reduction in crop productivity (9). Therefore, collective efforts are necessary to reduce the adverse effects of waterlogging stress on crop production (10). Even though much has been achieved by the genetic improvement of crop cultivars and cultivation measures that mitigate waterlogging, the role played by rhizosphere microorganisms in plant resistance to waterlogging has been limitedly studied.

Several studies have reported microorganisms with varied positive effects growing near stressed plants (11, 12). For example, beneficial bacteria, such as *Bacillus* containing 1-aminocyclopropane-1-carboxylic acid deaminase, can reduce stress-induced ethylene content, thereby protecting plants from waterlogging (5, 6). Moreover, some bacteria, such as *Bacillus thuringiensis*, can synthesize indole-3-acetic acid in the rhizosphere (13), which indirectly helps to reduce the damage caused to plants by waterlogging (14). An anaerobic condition caused by waterlogging stress might change the structure of the soil microbial community, which might further affect the composition of the ecosystem (15, 16). Therefore, exploring the response of rhizosphere microbes to waterlogging stress can provide a basis for using beneficial microbes to improve the resistance of soybean (17).

High-throughput sequencing of partial 16S rRNA is the most common method used to analyze the microbial community due to its low cost (18). Often only V1 to V3, V3 to V5, or V4 of the nine variable regions in the 16S rRNA gene (called V1 to V9) have been interrogated (19–21). However, the taxonomic classification as well as the abundance and diversity of operational taxonomic units (OTUs) are affected by variable region selection (22, 23). Moreover, the accuracy and sensitivity of taxonomic discrimination and estimates of taxon abundance are significantly influenced by sequence read length and primer selection (24, 25). The short-read approach by second-generation sequencing was found to be affected by the variable region bias and could not provide valid information beyond the genus level. This resulted in an inaccurate classification of sequences, especially in the environmental samples (26, 27). Low-resolution classification not only limits the accuracy of microbial ecological function inference and host metabolic reconstruction but also affects the appropriate identification of bacterial strains in subsequent experiments and transformation studies (27).

Pacific Bioscience (PacBio) has developed a long-read sequencing technology that can complete full-length 16S rRNA gene (V1 to V9) sequencing at comparatively high throughput (24, 28). Moreover, the initial high intrinsic error rate has been improved by the circularized library templates combined with highly processed polymerases that allow for the "circular consensus sequence" (CCS) read with sufficiently high quality (19). Recently, a new full-length sequencing method called LoopSeq, with high accuracy and less expensive prices, has started to be used in some research (29). LoopSeq can eliminate bias caused by PCR and can sequence molecules with very low abundance. In this process, quantitative PCR (qPCR) is performed to first quantify the DNA

**TABLE 1** Related attributes of different sequencing platforms

| Sequencing characteristics | V4 | LoopSeq | PacBio |
|---|---|---|---|
| Cloning required | No | No | No |
| Avg sequence time | 8 h | 2.5 h | 2 h/SMRT cell |
| Avg read length | ~250 bp | ~1,500 bp | 1,000–1,500 bp |
| Read technology | Short-read technology | Long-read technology | Long-read technology |
| Sequencing variable region | V4 | V1–V9 | V1–V9 |
| Stitching during sequencing | Yes | No | No |
| Error rates | High | Lower | Medium |
| Eliminate PCR bias | No | Yes | Yes |
| Need stitching | Yes | No | No |
| Approximate cost per Mb | US$0.11 | US$0.245 | US$2.50 |

before pooling the DNA into a single reaction. The sequencing procedure lowers the effects of microbial absolute abundance among different samples, and, therefore, a more accurate picture of the species in the sample is reported. Moreover, each 16S molecule is barcoded before clustering and assembling into a single long read. This results in overlapping of each base position with multiple short-read sequences and allows for consensus calling to determine the true call independent of sequencing errors (<0.005%) (30). However, how different sequencing methods affect environmental samples, which refers to the response of rhizosphere microbes in different types of soil to waterlogging stress in this study, has not been reported.

In the current study, the effects of waterlogging stress on soybean rhizosphere bacterial structure in two types of soil were explored using partial 16S rRNA and full-length 16S gene sequencing with LoopSeq and PacBio. We hypothesized that (i) waterlogging will decrease the bacterial diversity in the soybean rhizosphere, (ii) the resolution would be similar between PacBio and LoopSeq and both will be significantly higher than partial sequencing, and (iii) specific microorganisms will be enriched in the soybean rhizosphere that may help soybeans resist waterlogging stress.

## RESULTS

The advantages and disadvantages of the three sequencing methods are presented in Table 1. In general, full-length sequencing had a high-throughput nature, which could eliminate PCR bias and cover more sequencing areas, while V4 sequencing was the least expensive (Table 1).

**Effects of waterlogging and soil types on bacterial diversity and community structure.** The sequencing methods, waterlogging, soil types, and their interactions significantly affected the alpha-diversity (Table S2 in the supplemental material). Higher alpha-diversity was obtained with V4 sequencing than with full-length sequencing. In more detail, the results from the three sequencing methods showed that waterlogging had no significant impact on the soybean rhizosphere bacterial diversity, except in the samples from acidic soil sequenced by LoopSeq (Dunn's $t$ tests, $n = 12$, $P > 0.05$) (Fig. 1A). Using the V4 sequencing method, we revealed that neutral soil had higher Shannon diversities than acidic soil (Dunn's $t$ tests, $n = 12$, $P < 0.05$) (Fig. 1B). Regarding beta-diversity, constrained principal-coordinate analysis (CPCoA) revealed that waterlogging and soil types had significant effects on rhizosphere microbial community structure in V4 and LoopSeq sequencing. However, from PacBio sequencing, only waterlogging had a significant effect on rhizosphere microbial community structure (permutational multivariate analysis of variance [PERMANOVA], $n = 12$, $P < 0.05$) (Fig. 1C and E and Table 2).

**Taxonomic comparison of different sequencing methods.** To determine whether differences in sequencing regions affected the assignment of sequences to different taxonomic levels, we analyzed the annotation proportion of 48 data sets for V4, LoopSeq, and PacBio each at the phylum, class, order, family, genus, and species levels. The results showed that the sequencing methods, soil types, and their interactions significantly affected the annotation proportion at low classification levels (such as on species level) (Table S2). In

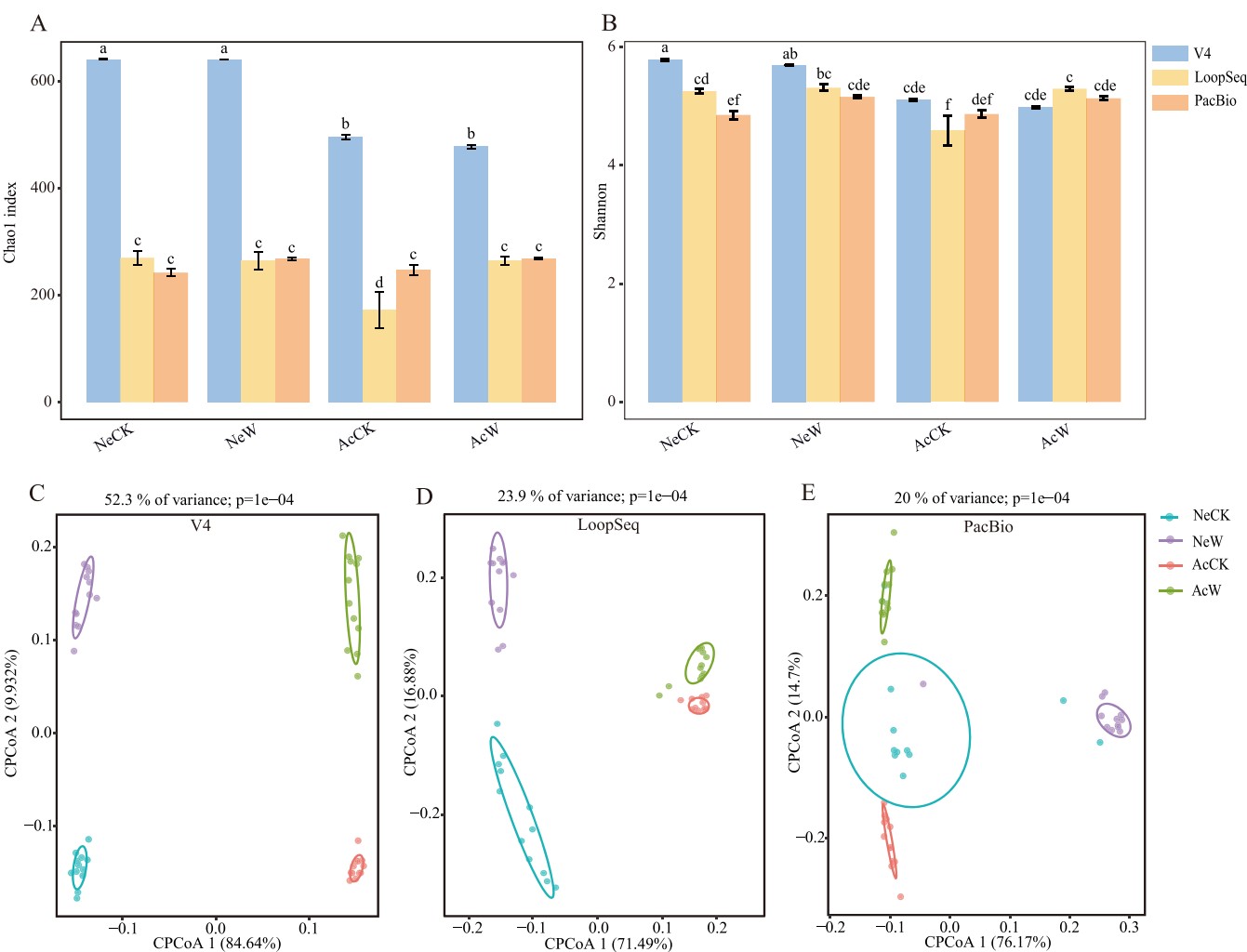

**FIG 1** (A, B) Effects of waterlogging and soil types on soybean rhizosphere soil bacterial Chao1(A) and Shannon index (B) with Illumina MiSeq, LoopSeq, and PacBio full-length sequencing methods (Dunn's *t* tests, *n* = 6, *P* < 0.05). (C to E) Constrained principal-coordinate analysis (CPCoA) of V4 (C), LoopSeq (D), and PacBio (E) data based on Bray-Curtis distance showing differences in rhizosphere bacterial community structure under waterlogging in neutral and acidic soil (PERMANOVA, *n* = 6, *P* < 0.05). Different letters indicate significant differences (*P* < 0.05). Ne, neutral soil; Ac, acidic soil; W, waterlogging; CK, without waterlogging; NeCK, soybean rhizosphere soil without waterlogging in neutral soil; NeW, soybean rhizosphere soil with waterlogging in neutral soil; AcCK, soybean rhizosphere soil without waterlogging in acidic soil; AcW, Soybean rhizosphere soil with waterlogging in acid soil; V4, Illumina MiSeq; LoopSeq, full-length Loop Genomics sequencing technology; PacBio, full-length PacBio single-molecule, real-time (SMRT) technology.

detail, the classification resolution on all the taxonomy levels of different sequencing methods was significantly different (*P* < 0.05) (Fig. 2A, C, and E; Fig. S1A, C, and E). At the phylum level, the proportion of assigned sequences was ranked as V4 > LoopSeq > PacBio (Fig. 2A), and at the class level, the proportion of assigned sequences was ranked as PacBio > V4 > LoopSeq (Fig. S1A). However, at other levels, the PacBio data sets had the highest proportion of assigned sequences (Fig. 2B and E; Fig. S1C and E). Venn analysis showed that the numbers of shared phyla, genera, and species were 16, 201, and 24, respectively, among the three different sequencing methods. Furthermore, the numbers of unique genera and species of LoopSeq and PacBio sequencing were significantly higher than those of V4 sequencing. The numbers of shared classes, orders, and families were 33, 58, and 115, respectively, among the three different sequencing methods (Fig. 2B, D, and F; Fig. S1B, D, and F).

**Rhizosphere microbial community structure of different sequencing methods at the phylum level.** Analysis at the phylum level showed that the effect of waterlogging on soybean rhizosphere soil microbial relative abundance in the two types of soil was different among the three sequencing methods. A total of 44 phyla were detected

**TABLE 2** Effects of waterlogging and soil types on bacterial community structure in soybean rhizosphere analyzed by permutational multivariate analysis of variance (PERMANOVA)

| Sequencing methods | Factor[a] | F | $R^2$ | P |
|---|---|---|---|---|
| $V_4$ | Soil | 111.7140 | 0.66882 | 0.001[b] |
| | Water | 10.2152 | 0.06116 | 0.002[c] |
| | Soil:water | 1.1018 | 0.00660 | 0.246 |
| | NeW vs NeCK | 7.0589 | 0.24292 | 0.001[b] |
| | AcW vs AcCK | 6.7174 | 0.23391 | 0.001[b] |
| LoopSeq | Soil | 17.5165 | 0.26413 | 0.001[b] |
| | Water | 2.6542 | 0.04002 | 0.012[d] |
| | Soil:water | 2.1465 | 0.03237 | 0.026[d] |
| | NeW vs NeCK | 2.5809 | 0.105 | 0.002[c] |
| | AcW vs AcCK | 3.0537 | 0.12189 | 0.001[b] |
| PacBio | Soil | 0.9707 | 0.01801 | 0.358 |
| | Water | 7.9983 | 0.14838 | 0.001[b] |
| | Soil:water | 0.9332 | 0.01731 | 0.386 |
| | NeW vs NeCK | 2.5963 | 0.10556 | 0.001[b] |
| | AcW vs AcCK | 7.9223 | 0.26476 | 0.001[b] |

[a]NeCK, soybean rhizosphere soil without waterlogging in neutral soil; NeW, soybean rhizosphere soil with waterlogging in neutral soil; AcCK, soybean rhizosphere soil without waterlogging in acidic soil; AcW, soybean rhizosphere soil with waterlogging in acid soil.
[b]Significant P value of <0.001.
[c]Significant P value of <0.01.
[d]Significant P value of <0.05.

by V4 sequencing, among which *Proteobacteria*, *Acidobacteria*, and *Chloroflexi* were dominant, with relative abundances ranging from 31.1% to 39.1%, from 15.0% to 21.9%, and from 5.9% to 7.3%, respectively. A total of 36 phyla were detected by LoopSeq sequencing, among which *Proteobacteria*, *Acidobacteria*, and *Actinobacteria* were the most pronounced, with relative abundances from 33.5% to 40.3%, from 18.4% to 25.7%, and from 7.6% to 10.8%, respectively. PacBio sequencing detected 29 phyla, among which *Proteobacteria*, *Planctomycetes*, and *Bacteroidetes* were dominant, with relative abundances from 30.7% to 35.4%, from 15.9% to 21.1%, and from 9.8% to 12.8%, respectively (Fig. S2). Waterlogging increased the relative abundance of *Firmicutes* and *Gemmatimonadetes* in the two types of soil, but the increase was significantly higher in the acidic than in the neutral soil for the three sequencing methods ($P < 0.05$) (Fig. S3J and K).

**Rhizosphere microbial community structure of different sequencing methods at the genus level.** Discrepancies among microbial community profiles represented by different sequencing methods were obvious at the genus level (Fig. 3). The first 30 genera with higher relative abundance were selected from the shared 201 genera from the three methods. Waterlogging was shown to increase the relative abundance of *Pirellula* in the two types of soil in only the PacBio sequencing method. However, the relative abundance of *Geobacter* was increased by waterlogging in all the sequencing methods.

**Rhizosphere microbial community structure of different sequencing methods at the OTU level.** Differential expression analysis of OTUs was also performed. In V4 sequencing, waterlogging increased 60 OTUs that were shared among both neutral and acidic soils ($P < 0.05$). These OTUs were classified at the genus level and were mainly composed of *Geobacter* (7 OTUs, 2.4%), *Nitrospira* (1 OTU, 0.7%), and *Anaeromyxobacter* (3 OTUs, 0.4%) (Fig. S4A). In LoopSeq sequencing, waterlogging increased 10 OTUs that were shared between neutral and acidic soils ($P < 0.05$). Classification of these OTUs at the genus level showed that most of them were *Oryzihumus* (1 OTU, 0.5%), *Massilia* (1 OTU, 0.18%), and *Acidothermus* (1 OTU, 0.16%) (Fig. S4B). Waterlogging increased the 9 OTUs in PacBio sequencing that were shared between the two types of soil ($P < 0.05$). These OTUs were classified at the genus level and were mainly made up of *Flaviolibacter* (1 OTU, 0.67%), *Ramlibacter* (1 OTU, 0.64%), and *Geobacter* (4 OTUs, 0.43%) (Fig. S4C). After assigning these OTUs to the genus, we did not find the coenriched microbial species in the three sequencing methods (Fig. S4D).

**Core microbiome analyses.** There were 18, 4, and 10 core OTUs in V4, LoopSeq, and PacBio sequencing, respectively (Table S3). In V4 sequencing, the core OTUs only

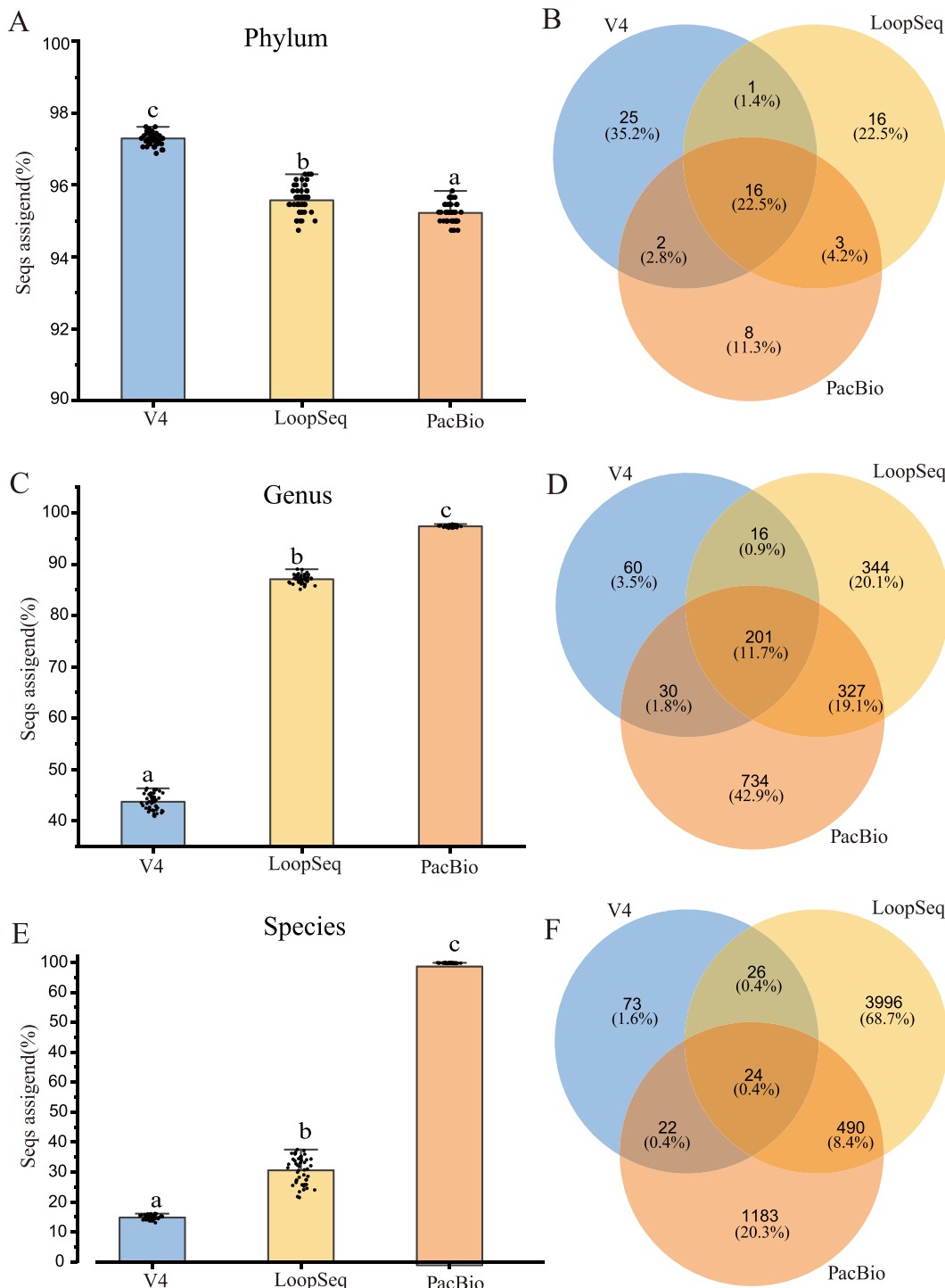

**FIG 2** Taxonomy profiles in different sequencing method data sets. (A, C, E) The proportion of annotation sequences from the V4 ($n = 48$, blue), LoopSeq ($n = 48$, yellow), and PacBio ($n = 48$, orange) data sets was determined by comparing the sequence with the SILVA database and is represented at the phylum (A), genus (C), and species (E) levels. (B, D, F) Venn diagram showing the numbers of unique and shared phyla (B), genera (D), and species (F) between the three sequencing methods. Blue denotes V4, yellow denotes LoopSeq, and orange denotes PacBio.

accounted for 0.69% of the total number of OTUs. Proteobacteria was the most abundant in the core rhizosphere microbial community and was composed of 8 OTUs and accounted for 5.2% of the average relative abundance. Other OTUs represented in the core rhizosphere microbiome were 5 *Acidobacteria* OTUs (1.56% relative abundance), 1 *Bacteroidetes* OTU (0.99% relative abundance), 1 *Firmicutes* OTU (0.61% relative abundance),

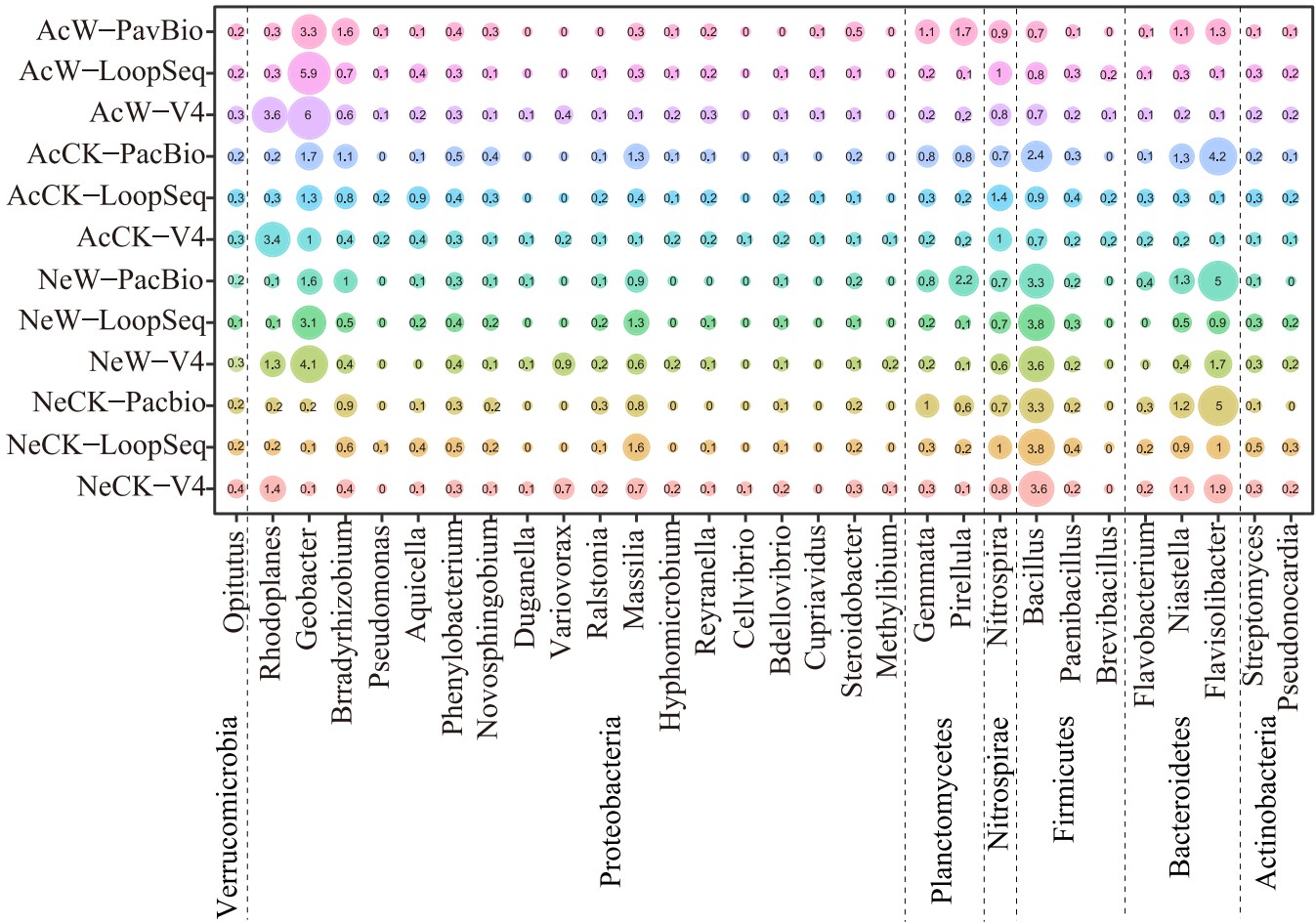

**FIG 3** Relative abundance analysis of common genera in three sequencing methods. The most abundant 30 genera were selected from shared 201 genera of the three sequencing methods. Color pairs denote samples of three sequencing methods in neutral or acidic soil with different waterlogging times. Bubble sizes indicate the averaged relative abundance of an individual genus across each treatment. The explanations of the abbreviations of different treatments were the same as in the legend to Fig. 1.

1 *Gemmatimonadetes* OTU (0.54% relative abundance), 1 *Nitrospirae* OTU (0.32% relative abundance), and 1 *Verrucomicrobia* OTU (0.32% relative abundance) (Fig. 4A). In LoopSeq sequencing, core OTUs only accounted for 0.061% of the total number of OTUs. This core rhizosphere microbiome consisted of 2 *Actinobacteria* OTUs (1.24% relative abundance) and 2 *Firmicutes* OTUs (1.12% relative abundance) (Fig. 4B). In PacBio sequencing, the core OTUs represented only 0.026% of the total number of OTUs. This core rhizosphere microbiome had 4 *Proteobacteria* OTUs that made up 2.09% of the mean relative abundance. Other OTUs in this core rhizosphere microbiome were 2 *Bacteroidetes* OTUs (1.18% relative abundance), 1 *Nitrospirae* OTU (0.36% relative abundance), and 1 *Planctomycetes* OTU (0.31% relative abundance) (Fig. 4C). Among them, OTU4 (*Bacillus*) and OTU4746 (*Bacillus*) were the core species shared by V4 and LoopSeq sequencing, while OTU17 (*Nitrospira*) and OTU22 (*Nitrospira*) were core species shared by V4 and PacBio sequencing.

**Modular analysis of the cooccurrence network.** Network modeling was applied to assess the composition and structure of microorganisms with different sequencing methods, and the OTUs with sequence numbers greater than five were screened as nodes. The networks of the V4, LoopSeq, and PacBio data sets were each divided into seven major modules. In V4, modules I, II, and III accounted for 35.85%, 32.76%, and 28.45% of the whole network, respectively (Fig. 5A). In LoopSeq, modules I and II accounted for 32.98% and 14.61% of the whole network, respectively (Fig. 5B), whereas in PacBio, modules I, II, III, and IV accounted for 28.82%, 22.94%, 20.59%, and 20% of the whole network, respectively (Fig. 5C).

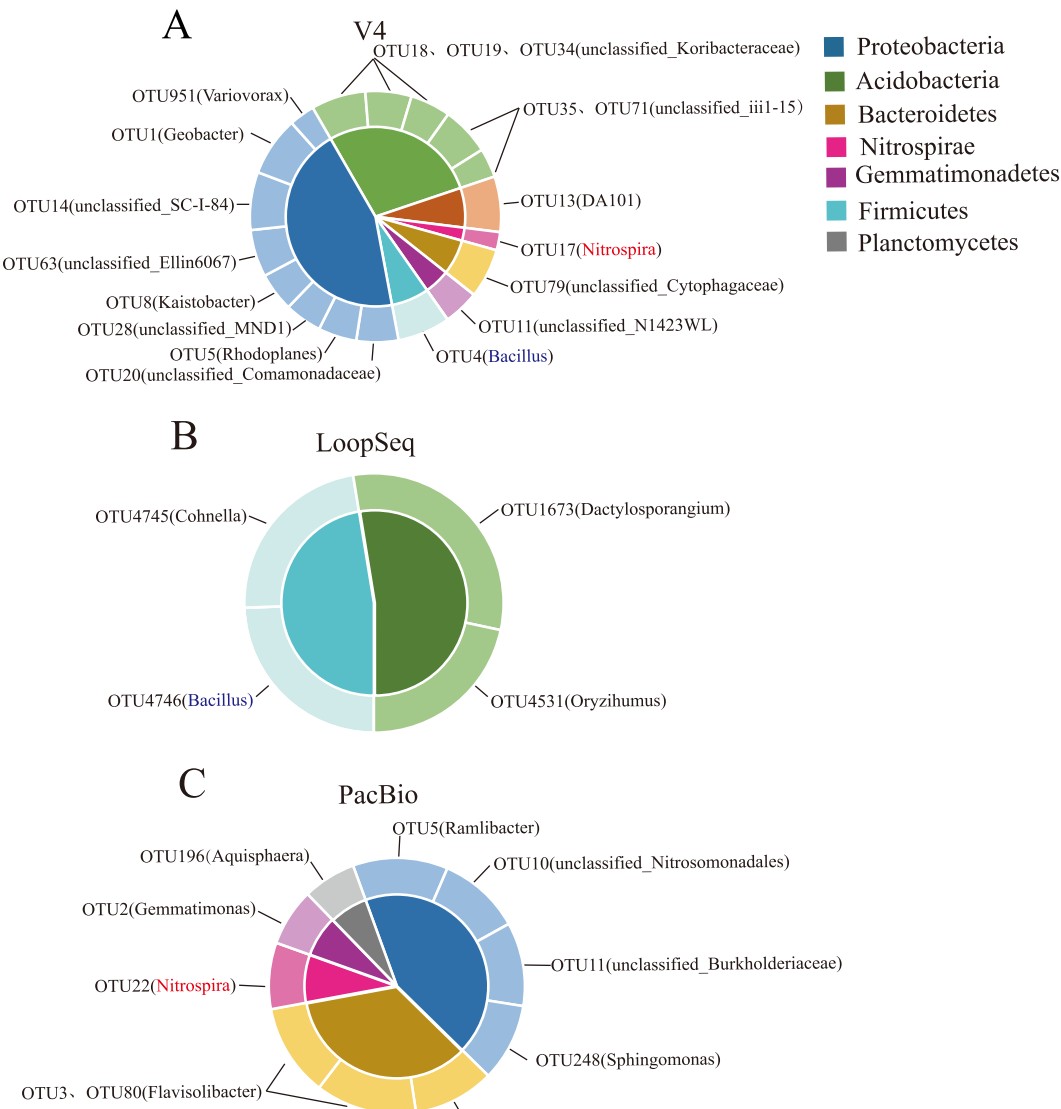

**FIG 4** Core microorganisms of the three sequencing methods. (A to C) Rhizosphere core microorganisms of the V4 (A), LoopSeq (B), and PacBio (C)sequencing methods. The different  parts inside the double pie chart represent the bacterial phyla of the soybean core microbiome. The different parts outside the double pie chart represent the OTU (genus) of the soybean core microbiome, and each OTU (genus) is assigned to the corresponding bacterial phyla. The size of the different double pie chart portions represents the percentage of phylum/genus relative abundance in all core microbial components.

To evaluate the differences in OTUs in different soils and under different waterlogging treatments using the three sequencing methods, we created a generalized linear model of the negative binomial distribution to analyze modules with high percentages. OTU numbers in the rhizospheres of the three sequencing methods under nonwaterlogging and acidic soil conditions were used as the controls to compare the enriched or depleted OTUs under the waterlogging and neutral soil conditions, respectively (Table S4).

The volcano plot of V4 sequencing showed that all OTUs of module II and module III had higher relative abundances in the waterlogging treatments than in the nonwaterlogging treatments, whereas all the OTUs of module I had lower relative abundances in the waterlogging treatments than in the nonwaterlogging treatments. The most OTUs of modules I, II, and III had higher relative abundances in neutral soil than in acidic soil. In LoopSeq sequencing, the 4 OTUs of module I had higher relative abundances in the waterlogging treatments than in nonwaterlogging treatments, whereas

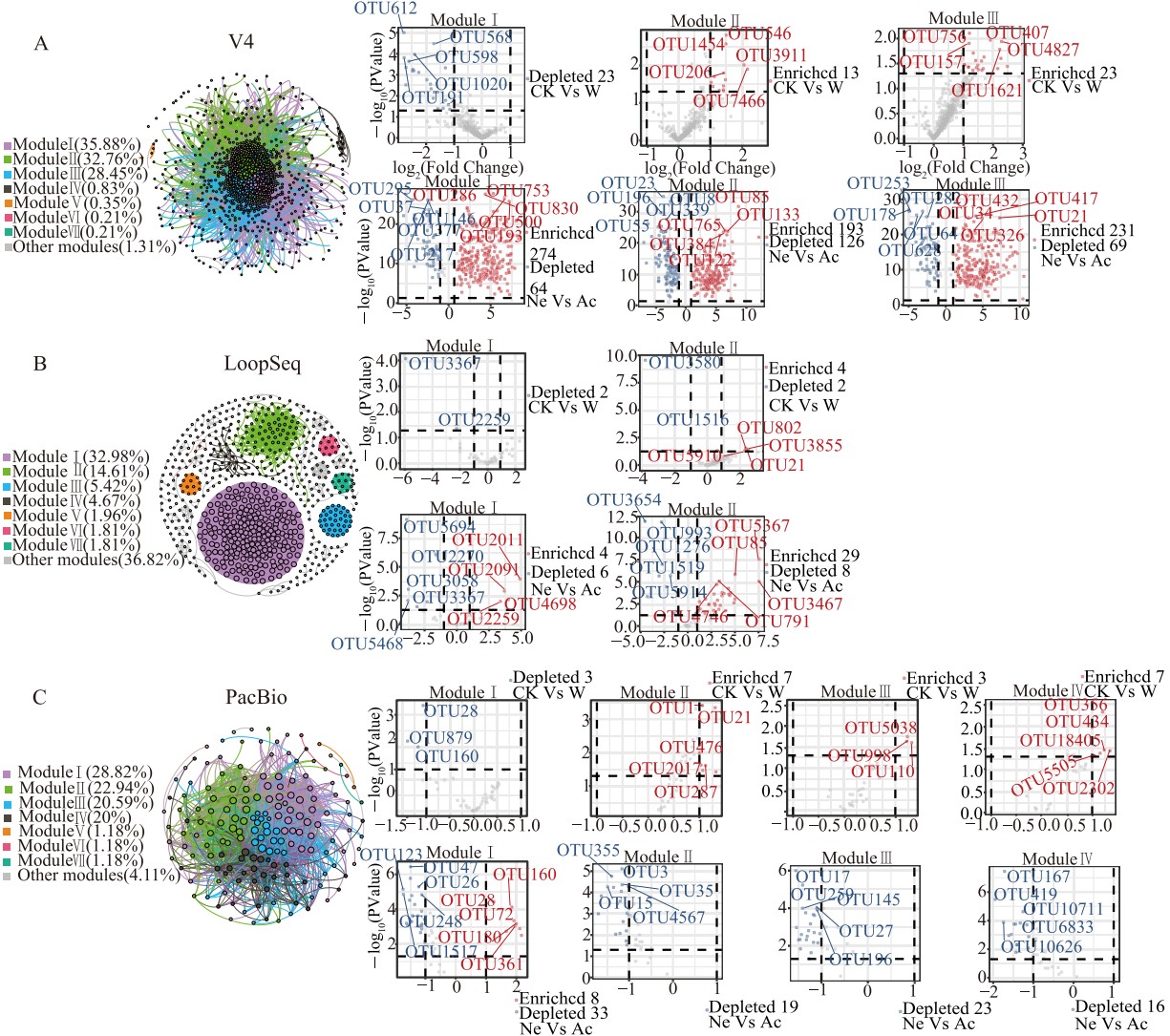

**FIG 5** (A to C) Network analysis reveals the symbiotic pattern between OTUs. The nodes are colored according to the modular type. The connections between nodes indicate strong and significant (Spearman's $r > 0.8$ or $r < -0.8$) ($P < 0.01$) correlation. The volcano map shows the amount of OTU enriched and depleted in neutral soil and after waterlogging in the modules of different sequencing methods, respectively. Data from V4 (A), LoopSeq (B), and PacBio (C) are shown. Violet denotes module I, green denotes module II, blue denotes module III, black denotes module IV, orange denotes module V, red denotes module VI, cyan denotes module VII, and gray denotes other modules.

the 2 OTUs of module I and all OTUs of module II had lower relative abundances in the waterlogging treatments. Most OTUs of module I and 4 OTUs of module II had higher relative abundances in neutral soil than in acidic soil. For PacBio sequencing, all OTUs of modules II, III, and IV had higher relative abundances in the waterlogging treatments than in the nonwaterlogging treatments. All OTUs of module I had lower relative abundances in the waterlogging treatments than in the nonwaterlogging treatments. The 8 OTUs of module I had higher relative abundances in neutral soils than in acidic soils, whereas most OTUs of module I and all OTUs of modules II, III, and IV had lower relative abundances in neutral soils than in acidic soils.

Nodes with high node degree, closeness centrality, and betweenness centrality were defined as the key species of the rhizosphere network (Table S5). In general, OTU7370 (*unclassified_Sphingobacteriales*), OTU405 (*unclassified_Verrucomicrobia*), OTU2287 (*unclassified_Ellin6513*), and OTU769 (*unclassified_SJA-28*) were identified as keystone species for V4 sequencing, while OTU1516 (*Solirubrobacter*), OTU5896 (*Conexibacter*), OTU791 (*Geobacter*), and OTU3146 (*Massilia*) were key species for the LoopSeq sequencing. For the PacBio

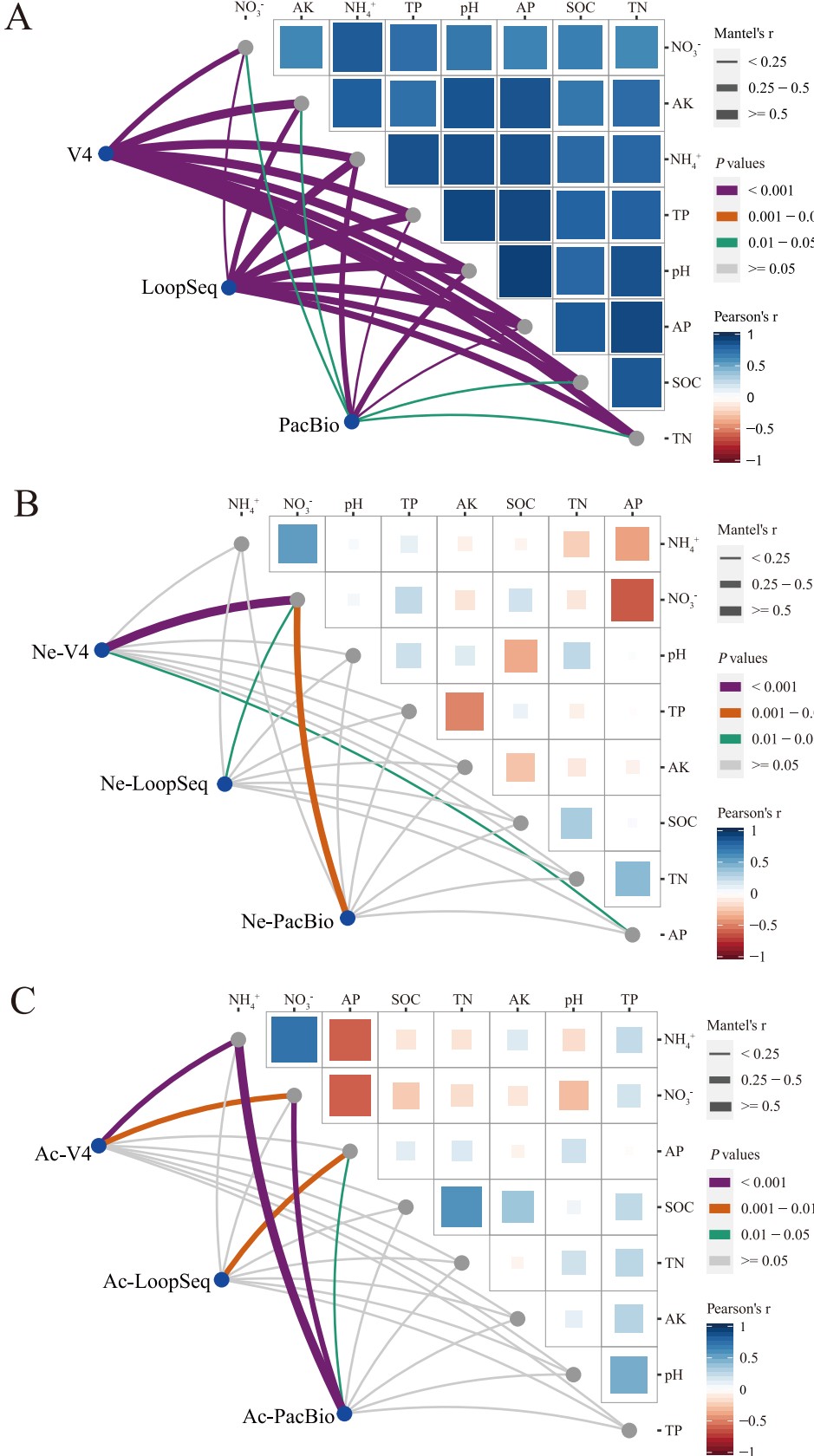

**FIG 6** (A to C) Paired comparison of environmental factors and microbial community with a color gradient denoting Pearson's correlation coefficient. A Spearman's correlation coefficient of >0 indicates positive

sequencing, the key species were OTU17 (*Gemmatimonas*), OTU18 (*Flavisolibacter*), OTU355 (*Aquisphaera*), and OTU10711 (*Algisphaera*).

**Environmental drivers of different sequencing methods.** The two-way analysis of variance (ANOVA) revealed that waterlogging, soil types, and their interaction significantly affected some soil physicochemical properties. For example, waterlogging significantly affected the Olsen P, $NH_4^+$, and $NO_3^-$, and their interaction significantly affected the $NO_3^-$ (Table S1). To determine the correlation between soil properties and microbial communities, we conducted a paired comparison of environmental factors. Our results showed that V4, LoopSeq, and PacBio sequencing had significant correlations with all soil physical and chemical factors (Fig. 6A). As the physical and chemical properties of the two soils were significantly different, we compared the relationship between the physical and chemical properties of the two soils and the microbial communities of the three sequencing methods, respectively. In neutral soil, the microbial structure sequenced by the V4 method was significantly correlated with available phosphorus (AP) ($P < 0.05$) and extremely significantly correlated with $NO_3^-$ ($P < 0.01$). The microbial structures sequenced by LoopSeq and PacBio were only significantly correlated with $NO_3^-$ ($P < 0.05$) (Fig. 6B). In acidic soil, the microbial structure sequenced by V4 was extremely significantly correlated with $NH_4^+$ and $NO_3^-$ ($P < 0.01$). The microbial structure sequenced by LoopSeq was only extremely significantly correlated with AP ($P < 0.01$). For PacBio, the microbial structure was significantly correlated with AP ($P < 0.05$) and extremely significantly correlated with $NH_4^+$ and $NO_3^-$ ($P < 0.01$) (Fig. 6C).

## DISCUSSION

In this study, we used three sequencing methods to evaluate the impact of waterlogging on the structure of soybean rhizosphere microbial communities in two types of soil. Our first hypothesis was not verified, as the results from the three sequencing methods showed that waterlogging had no significant impact on the bacterial alpha-diversity (Fig. 1A and B). However, V4 sequencing, but not full-length sequencing, showed significantly more alpha-diversity in neutral soil than in acidic soil. A reasonable conclusion might be that if the researchers had relied on V4 sequencing, they would have concluded (probably incorrectly) that soil type clearly influenced bacterial alpha-diversity, whereas with a better sequencing method (PacBio or LoopSeq), they would have accurately found no (or context-dependent) differences in soil types. For the other two hypotheses, they were fully verified, where the resolutions of PacBio and LoopSeq were significantly higher than partial sequencing, and some beneficial microorganisms, such as *Geobacter*, were enriched in the soybean rhizosphere that may help soybeans resist waterlogging stress.

Based on CPCoA, the results from all the sequencing methods showed that waterlogging significantly affected the rhizosphere bacterial community structure (Fig. 1C to E), and this was in agreement with the arguments put forward by Evans and Wallenstein (17). When the soil is waterlogged, the oxygen content of the soil sharply decreases, which reduces the respiration rate and activity of soil microorganisms. This in turn leads to the expected changes in microbial community structure (31–34). Furthermore, changes in crop root exudates induced by waterlogging also directly affect rhizosphere microbial community structure (35). Crops suffering from waterlogging stress affect the underground carbon input (36, 37), which then affects the rhizosphere microbiome (38). We selected acidic soil and neutral soil in this study, with an expectation that soil could significantly affect the rhizosphere microbial community

**FIG 6** Legend (Continued)
correlation, and a coefficient of <0 indicates a negative correlation. Effects of environmental factors in two types of soil (A), neutral soil (B), and acidic soil (C) on the microbial communities of the three sequencing methods are shown. The edge width corresponds to the distance dependence of Mantel's *r* statistic, and statistical significance based on 9,999 permutations represents edge color. Mantel's *r* size indicates the strength of the correlation. The color of the connecting line indicates the correlation between different sequencing methods and environmental factors.

structure irrespective of the sequencing methods. However, we found that soil type was not a significant factor driving changes in microbial community structure in PacBio sequencing. This could be attributed to long sequencing, which could lead to the reconstruction of phylogeny and thus affect the similarities or differences of microbial communities (39).

Our results showed that full-length sequencing (except at the phylum level) had a higher classification resolution (Fig. 3; Fig. S1; Table S2). This was anticipated, as full-length read sequence has been shown to provide a higher phylogenetic classification resolution (40). When sequencing with different variable regions, almost all the sequences of V1 to V9 were annotated to the species level compared with other variable regions (25, 41). Because full-length sequencing covers most of the target genes, it has a high-resolution capacity to discriminate many phylogenetic closely related taxa (42, 43). However, the resolution of LoopSeq was lower than PacBio, which could be due to differences in the sequencing platform. LoopSeq uses the Illumina platform for full-length sequencing. PacBio's CCS library can improve the accuracy by sequencing a single fragment for multiple rounds, leading to a more accurate species classification (41). Similar to previous studies (44), we found that the classification of microbial groups is affected by a smaller 16S amplicon. The V4 data sets suffered from this bias, which further supported the use of longer readings for microbial ecological analysis (45).

To determine whether the high species classification resolution of full-length sequencing could help in identifying more microorganisms related to waterlogging resistance, we compared the microbial community structure at the genus level using three sequencing methods. Our results revealed that the effect of waterlogging on rhizosphere soil bacteria was different across the sequencing methods. For example, a significant difference in *Variovorax* was only detected in V4 sequencing and has been previously reported to manipulate plant ethylene levels to balance normal root development (46), thus alleviating the harm of waterlogging (5). The increased relative abundance of *Pirellula*, which plays an important role in nitrogen cycling, was only found in PacBio sequencing (47, 48). These results indicated that there were differences in the information about the recruited bacteria detected by different sequencing methods. Due to the high resolution of full-length sequencing, we may conclude that the use of V4 sequencing may not give us very accurate information about the recruited microorganisms, which might affect our screening of waterlogging tolerance-related microorganisms. However, we still found some common features in some microbial genera that respond to waterlogging among the three sequencing methods (Fig. 4). The increased abundance of *Geobacter* in waterlogging stress was detected in the three sequencing methods. *Geobacter* plays an important role in plant nitrogen fixation (49, 50) and can secrete fulvic acid and participate in plant electron transfer (51–53), which may be related to electrical signals mediated by plant potassium channels (54). The hypoxic environment under waterlogging stress results in a sharp decline in microorganisms involved in the nitrification reaction. This inhibits the activity of the nitrifying community, leading to increased nitrogen loss (55). The enrichment of anaerobic bacteria (such as *Geobacter*) may fix more nitrogen, thereby allowing plants to grow healthily. However, the extent to which and the mechanism through which *Geobacter* improves the adaptability of a plant to waterlogging stresses remains unknown and needs to be explored in the future.

At the OTU level, the effect of waterlogging on the two soils was also different among the three sequencing methods. The OTUs that significantly changed in both types of soil after waterlogging belonged mainly to *Geobacter*, *Anaeromyxobacter*, and *Nitrospira* in V4, *Oryzihumus*, *Massilia*, and *Acidothermus* in LoopSeq, and *Flavisolibacter*, *Ramlibacter*, and *Geobacter* in PacBio, respectively. Among them, *Geobacter* was observed in both V4 and PacBio sequencing methods, which was consistent with analysis at the genus level. Previous studies have shown that *Geobacter* and *Nitrospira* are related to microbial nitrogen fixation (50, 56). These genera are reductive microorganisms (57), which can use a wide range of carbon and/or electron donors to participate in metabolic pathways. The broad metabolic diversity of microorganisms was considered to be advantageous, particularly at times of

nutrient scarcity (58). However, long sequencing revealed more differences in OTUs that have other functions. Some microorganisms identified by LoopSeq and PacBio are related to phosphorus cycling and high soil fertility. For instance, *Massilia* could help the turnover of root exudates, such as amino acids, sucrose, and fatty acids, and provide phosphorus solution to plants (59, 60). *Acidothermus* can decompose organic matter and utilize carbon sources, thus enriching the soil organic matter content (61). *Flavisolibacter* has the effect of dissolving phosphorus in the soil (62, 63). Based on these results, long sequencing might detect more microbial information related to waterlogging tolerance.

Core microorganisms with different functions are involved in the coordination and organization of plant-microbe interactions (64). Three sequencing methods resulted in different species of the core microbiome, which mainly included *Nitrospira*, *Geobacter*, *Variovorax*, and *Bacillus* in V4 sequencing, *Bacillus* and *Dactylosporangium* in LoopSeq sequencing, and *Nitrospira*, *Flavisolibacter*, *Gemmatimonadetes*, and *Ramlibacter* in PacBio sequencing (Fig. 5). In particular, *Nitrospira* was the core microorganism shared by V4 and PacBio sequencing, while *Bacillus* was the core microorganism shared by V4 and LoopSeq sequencing. *Nitrospira* is the most common genus affecting soil nitrogen metabolism (65–67). *Bacillus* can utilize multiple electron donors or collectors to enrich nutrients (57) and maintain normal root growth (46). Additionally, *Geobacter*, which is related to nitrogen fixation (50, 56), and *Flavisolibacter*, which can dissolve soil phosphorus (62, 63), were the core microorganisms for V4 and PacBio sequencing, respectively. These core microorganisms might help plants resist waterlogging stress through different nutrient cycling or recruit other beneficial microorganisms to resist the effects of waterlogging together with plants. Nevertheless, whether the core microorganisms that we discovered could establish a defense mechanism against waterlogging damage with soybeans is still unclear. To address these issues, the use of high-throughput cultivation and identification of microbes (68) as well as synthetic communities (69) can help explore the extent to which these recruited microorganisms contribute to soybean resistance to waterlogging stress.

Cooccurrence patterns are ubiquitous in nature and are particularly involved in the analysis of microbial community structure. Network cooccurrence analysis can provide an in-depth and unique perspective for understanding microbial interactions and ecosystem assembly rules rather than simple species diversity and composition (70–72). Network modularity may reflect collaborative relationships, competitive interactions, and niche differentiation, which leads to nonrandom patterns of interaction and affects the complexity of the ecological network (73). Dividing the network into modules helps to clarify different node groups that perform different functions (74). For example, the main modules with a high percentage in V4 (except module I) and LoopSeq sequencing are enriched with some microorganisms related to nitrogen cycling (e.g., *Mucilaginibacter*, *Candidatus Solibacter*, *Candidatus Koribacter*, *Geobacter*, and *Bacillus*) after waterlogging (49, 75). This agrees with previous studies where the nitrogen-fixing microorganisms might be enriched in the waterlogging soil (76, 77). Moreover, the main modules with a high percentage of LoopSeq and PacBio sequencing are enriched with some microorganisms related to phosphorus cycling (e.g., *Massilia* and *Flavisolibacter*) after waterlogging (62, 63). This showed that waterlogging can selectively increase or decrease part of the microbial abundance related to nitrogen cycling. However, the functions of depleted microorganisms in the main modules of LoopSeq and PacBio sequencing have not been reported. It is worth noting that the conclusions we obtained using network analysis were speculative and cannot be taken as definitive information. This is because networks provide a valuable tool, but they are best seen as hypothesis generators rather than solid conclusions. This is because these methods only infer ecological associations. Furthermore, the choice of network analysis method and network size can affect the accuracy and significance of the network, even when assumptions are met (78, 79).

Compared with acidic soil, the microorganisms related to nitrogen fixation (e.g., *Geobacter*, *Nitrospira*, *Candidatus Koribacter*, and *Candidatus Solibacter*) in the main modules of the network are enriched in neutral soil (49, 56, 75, 80, 81). This might indicate that

waterlogging is less harmful to neutral soils than acidic soils, at least on the level of microbial functions. However, microorganisms related to phosphorus cycling (e.g., *Flavisolibacter*, *Massilia*, and *Gemmatimonas*) were depleted in neutral soils (62, 63, 65). In this study, acidic soil had lower phosphorus content than neutral soil. A previous study showed that when P availability in soil is low, the enrichment of inorganic phosphate-solubilizing bacteria could efficiently transform immobilized P into bioavailable P with high phosphatase activities (82). Moreover, the keystone species in the rhizosphere varied among V4, LoopSeq, and PacBio sequencing, which might be a key determinant of the composition of other communities in the rhizosphere of plants (83).

To determine environmental factors affecting the microbial communities of three sequencing methods in different soils, we performed Pearson's correlation coefficient analysis using all samples from both types of soil. The results showed that all the environmental factors affected the microbial communities in both types of soil in the three sequencing methods. This might have been caused by the soil heterogeneity between neutral and acid soils (84). In neutral soil, $NO_3^-$ was the main environmental factor that affected the microbial community in all sequencing methods. Previous studies have shown that some period after waterlogging, the soil nitrogen form is still dominated by $NO_3^-$, which could be transported into the host by microorganisms (85, 86). For the acidic soil, $NH_4^+$ and $NO_3^-$ were the major environmental factors that affected the microbial community in V4 and PacBio sequencing. Total phosphorus (TP) affected the microbial community in LoopSeq and PacBio sequencing methods. These results were in line with the network module association analysis, which showed that phosphorus-related microorganisms were enriched in acidic soil. It has been previously reported that soil microbial community structure is significantly affected by soil phosphorus content (87). From these results, the impact of environmental factors on the microbial community was different among the three sequencing methods and with the soil type.

The nonnegligible difference of bacterial diversity, comparisons at the level of bacterial phyla and genera, core microorganisms, network cooccurrence analysis, and correlation with environmental factors was in line with that the sequencing bias should be taken into account by different sequencing methods for microbial community analysis (88, 89). Our results are consistent with previous findings that showed that while short-read sequencing is effective in microbiome analyses at higher taxonomic levels (e.g., phylum level), LoopSeq and PacBio analyses show greater power to delve deeper into taxonomic capabilities (24, 30). We cannot deny that the conclusions or predictions of previous studies based on V4 sequencing may be indicative of importance, but, at the least, in this study, we show that both full-length sequencing methods showed similar results and identified more functional microorganisms than V4 sequencing.

**Conclusion.** In summary, this study explored the effects of waterlogging on soybean rhizosphere microbial communities in different soils using three sequencing methods. Our results showed that full-length sequencing had a higher resolution than partial sequencing. Waterlogging had a significant impact on the rhizosphere microbial community structure, while only two sequencing methods (V4 and LoopSeq sequencing) showed that soil type could also significantly influence microbial community structure. Furthermore, both LoopSeq and PacBio detected that waterlogging enriched microorganisms related to phosphorus cycling, such as *Flavisolibacter* and *Massilia*. Core microorganisms and network modularity analysis further revealed that enriched different species might play central roles in maintaining the stability of bacterial community structure and ecological functions. Together, our study not only explored the role of microorganisms enriched at the rhizosphere level under waterlogging treatment in assisting soybeans to resist stress but also showed that LoopSeq sequencing is a less expensive and more convenient method for full-length sequencing by comparing different sequencing methods.

## MATERIALS AND METHODS

**Soil and soybean material.** Neutral and acidic soils were collected at the surface layer with a depth of 10 cm from Yingde County (113°40′N, 24°18′E) and Suixi County (110°25′N, 21°32′E), Guangdong Province,

China, respectively. Then, 500 kg of each type of soil were air dried for 5 days and sieved through 2-mm mesh to remove impurities before planting the soybeans. The neutral and acidic soil were classified as Kanhaplohumults and Paleustults, respectively, according to United States Department of Agriculture (USDA) soil taxonomy. Two soybean varieties (i.e., Qihuang34 and Jidou17) that are widely grown in central and southern China were used in this study. Soil physicochemical properties and rhizosphere microorganisms of both varieties were mixed as replicates for the subsequent analysis (90, 91).

**Experimental design.** We conducted a random pot experiment in the greenhouse at the Agricultural College of South China Agricultural University, Guangzhou, China. The experiment was a completely randomized block design. In total, 48 pots of soybeans with or without waterlogging treatment in two types of acidic soil (two soil types × two waterlogging treatments = four treatments) were used. For each treatment, there were 12 replicate pots (6 replicates for each variety) with 4 seedlings per replicate. Each pot (top diameter of 13.8 cm, bottom diameter of 10.4 cm, and height of 12.2 cm) used in this study contained about 2.5 kg of air-dried soil. Eight strong and full soybean seeds with similar shapes were sown in each pot. The soybean growth process was carried out in a greenhouse with controllable conditions (temperatures of 26 to 32°C in the daytime and 15 to 21°C in the nighttime). After 6 days of emergence, 3 healthy soybean plants with the same growth were left. Waterlogging stress examination was performed on soybeans in the V2 stage (the period of ternary compound leaf expansion) in order to explore three sequencing methods for examining waterlogging-affected soybean rhizosphere microbes. For waterlogging treatment, water was added to the pots up to 4 to 6 cm above the soil, and more water was added to the soil twice a day for 3 days to ensure the water level. For control plants, the water content was left in an ideal environment.

**Soil sampling.** After 3 days of waterlogging treatment, the bulk soil was removed from the root by manually shaking, and then the entire root of all three soybean plants in each pot was transferred into a 50-mL centrifuge tube filled with phosphate-buffered saline (PBS) to collect the rhizosphere soil (defined as the soil that adheres to the root). After that, the centrifuge tubes were placed on a shaker (120 rpm/min at 25°C) for 20 min and then centrifuged for 10 min (6,000 × $g$, 4°C). Five grams of deposited rhizosphere soil was collected from each sample and placed in a sterilization centrifuge tube for storage at −80°C for DNA extraction. The remaining rhizosphere soil was stored at 4°C prior to determination of soil physical and chemical properties.

**Analysis of soil properties.** After sampling for 1 week, the soil was air dried for 5 days and sieved through 2-mm mesh to remove plant residues. Then, soil physical and chemical properties were measured according to previous studies (92, 93). In general, soil pH was determined using a pH meter (FE20-FiveEasypH, Mettler Toledo, Germany) in soil water suspension (5:1 water-to-soil ratio). Total nitrogen (TN) was determined using a UV spectrophotometer (UV-1800, Suzhou, China). Total potassium (TK) in soil was measured using a flame atomic absorption spectrometer (AA-7000, Shimadzu, Japan). Soil organic carbon (SOC) content was assessed using a TOC-5000A analyzer (Shimadzu, Kyoto, Japan). The content of $NH_4^+$, $NO_3^-$, TP, and Olsen P in soil was determined by a continuous flow analytical system (Skalar San$^{++}$, Netherlands). The effects of waterlogging on soil chemical properties are summarized in Table S1 in the supplemental material.

**DNA extraction from soil samples and sequencing process.** Total soil DNA was extracted using a Fast DNA Spin kit for soil (MP Biomedicals, Santa Ana, CA) following the manufacturer's recommendations. The DNA was eluted with 80 $\mu$L of water and analyzed by Nanodrop 2000 spectrophotometry. Primers 515F (5′-GTGCCAGCMGCCGCGGTAA-3′) and 806R (5′-GGACTACHVGGGTTCTAAT-3′) with variable 12-bp barcode sequences were used to amplify the V4 region of the 16S rRNA gene (94). Primers 27F (5′-AGRGTTY GATYMTGGCTCAG-3′) and 1492R (5′-RGYTACCTTGTTACGACTT-3′) were used for full-length (V1 to V9) 16S rRNA gene amplification (LoopSeq and PacBio sequencing) (95). The qPCR system included 22.5 $\mu$L of PCR SuperMix, 1.0 $\mu$L of positive primer, 1.0 $\mu$L of reverse primer, 10 ng of template DNA, and double-distilled water (ddH$_2$O) supplemented to 25 $\mu$L. The amplification program was 1 cycle of 95°C for 60 s, 28 cycles of 95°C for 60 s, annealing at 58°C for 60 s, and primer extension at 72°C for 2 min and finally 1 cycle of 72°C for 10 min. Both V4 and full-length sequencing were performed according to the company of the HUADA BIG's standard procedures. An Illumina MiSeq platform was used to sequence the V4 amplicon (reagent kit v.3; Illumina). Full-length sequencing of PacBio was completed on the PacBio RS II platform.

A LoopSeq 16S microorganism 24-plex kit (Loop Genomics, San Jose, CA, USA) was used to analyze the microbial genome of rhizosphere soil. The unique molecular markers of a single 16S gene used in LoopSeq sequencing were distributed in the whole gene, and then the full-length 16S gene was recombined through short reading and sequencing on an Illumina platform. Briefly, 10 ng of DNA from different rhizosphere soil samples was used to build a sequencing library. The raw data were collected on an Illumina NextSeq, with generated FASTQ files (96). All raw data from V4 16S sequencing, LoopSeq sequencing, and PacBio sequencing were deposited in the National Microbiology Data Center (NMDC) under accession numbers NMDC10017771, NMDC10017785, and NMDC10017787, respectively.

**Data analysis.** For V4 16S and LoopSeq sequencing, the raw FASTQ sequence file was processed by QIIME 2. In brief, the divisive amplicon denoising algorithm 2 (DADA2) in the QIIME 2 plugin was used to obtain OTUs, which detected and corrected amplicon errors and filtered out the potential base error and chimeric sequences (30, 97). All the raw sequences were filtered, trimmed, and dereplicated. The representative sequence generated after denoising was based on sklearn's naive Bayes classifier for bacterial classification on the SILVA 16S full-length database (97, 98). For PacBio 16S sequencing, the raw sequence files were processed using single-molecule, real-time (SMRT) Link software version 5.1.0.26412 (Pacific Biosciences). The OTUs were clustered using the UPARSE algorithm (99), and parameters were used to tune the full-length sequencing. The OTUs were iteratively classified according to the latest non-redundant small subunit SILVA using the RDP classifier at a 99% cutoff. In order for amplicon sequence variants (ASVs) produced by the DADA2 algorithm to be consistent with OTUs produced by the UPARSE

algorithm and to be more concise in the description below, we replaced the ASVs with OTUs. Archaea information was removed from the sequence for subsequent analysis. In total, 1,716,803, 168,704, and 214,645 bacterial 16S rRNA high-quality reads were obtained from 48 samples, with an average of 35,766, 3,551, and 4,759 reads per sample after rarefied for the three sequencing methods, respectively. These read were sorted into 7,673, 6,494, and 38,106 OTUs of the three sequencing methods, respectively, for subsequent analysis.

**Statistics analyses.** Using the "vegan" package in R V3.6.3, constrained principal-coordinate analysis (CPCoA) based on the UniFrac distances, an Adonis test (PERMANOVA), and the Mantel test were performed (100). A three-way analysis of variance (ANOVA) and Dunn's multiple comparison with Bonferroni correction were used to identify significant differences in classification resolution, bacterial alpha-diversity, and the relative abundances of bacterial phyla and genera among all treatments. Moreover, a two-way ANOVA and multiple-comparisons testing were used to identify the significant differences in soil chemical properties (101). With each sequencing method, the relative abundance of OTUs in different treatments was determined using the "DESeq2" package with the Benjamini-Hochberg correction (DESeq2, $n = 12$, $P < 0.05$). Origin was used for the visualization of bar graphs of the classification resolution of different sequencing methods. A Venn diagram was used to show the numbers of unique and shared microbial species on different taxonomic levels of different treatments with the three sequencing methods (102).

The core bacteria, which contain a list of OTUs observed in 60% of V4, LoopSeq, and PacBio, were obtained using Microbiome Analyst (103). Additionally, we constructed three cooccurrence networks to analyze the correlations between OTUs in different sequencing methods (sequence number of >5). The "psych" package in R was used to calculated Spearman's rank correlation and $P$ value, and Gephi was used for visualization (104). Nodes were colored according to different modules. It is generally considered that nodes with a high degree, closeness centrality, and betweenness centrality values are the key species (105). To identify the OTUs that were significantly different for the network module between neutral soils and acidic soils as well as rhizosphere soil with and without waterlogging, we established a generalized linear model of the negative binomial distribution for differential OTU relative abundance analyses (98). Moreover, Spearman correlations in R were used to analyze the relationship between microbial communities and environmental factors in all the samples and the samples in two types of soil, respectively (106).

**Data availability.** All raw data from V4 16S sequencing, LoopSeq sequencing, and PacBio sequencing were deposited in the NMDC under accession numbers NMDC10017771, NMDC10017785, and NMDC10017787, respectively.

## SUPPLEMENTAL MATERIAL

Supplemental material is available online only.

**SUPPLEMENTAL FILE 1**, PDF file, 0.6 MB.

## ACKNOWLEDGMENTS

This work was financially supported by the National Key R&D Program of China (grant number 2018YFD1000903) and Guangzhou Science and Technology Innovation Development Funding (202102020068). We are very grateful to Xu Ran and Zhang Mengchen for providing the soybean varieties.

T.L. and H.N. conceived and designed the experiments. T.Y., T.L., L.C., Q.L., and S.W. performed the experiments. Y.Z., H.Z., and M.T. completed the library preparation and sequencing. T.Y., T.L., and H.N. analyzed the data and wrote the paper.

We declare no conflicts of interest.

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
