## [Reviewer comments · Microbiology Spectrum]

Microbiology Spectrum

Effects of waterlogging on soybean rhizosphere bacterial community using V4, LoopSeq and PacBio 16S rRNA sequence

Yu Taobing, Cheng Lang, Liu Qi, Wang Shasha, Zhou Yuan, Zhong Hongbin, Tang Meifang, Nian Hai, and Tengxiang Lian

Corresponding Author(s): Tengxiang Lian, South China Agricultural University

Review Timeline:

Submission Date:	October 26, 2021
Editorial Decision:	December 17, 2021
Revision Received:	December 27, 2021
Accepted:	January 16, 2022

Editor: Cheng Gao

Reviewer(s): Disclosure of reviewer identity is with reference to reviewer comments included in decision letter(s). The following individuals involved in review of your submission have agreed to reveal their identity: Glade Dlott (Reviewer #2)

Transaction Report:

DOI: <https://doi.org/10.1128/Spectrum.02011-21>

December 17, 2021

Dr. Lian Tengxiang
South China Agricultural University
Guangzhou
China

Re: Spectrum02011-21 (**Effects of waterlogging on soybean rhizosphere bacterial community using V4, LoopSeq and PacBio 16S rRNA sequence**)

Dear Dr. Lian Tengxiang:

Thank you for submitting your manuscript to Microbiology Spectrum. We now received two extensive reviews. Both reviewers highlight the significance and advance of this research, but also raise important suggestions for you to prepare the revised manuscript. Please pay more attention to the rationale of the hypothesis, details in experiment design, sampling, measure of soil parameters, bioinformatic (DADA2) and statistics (network), information of sequencing, archaea, and the effect of plant variety on the microbial community. Finally, the suggestion of reviewer#2 on the framework of the manuscript is for your consideration.

Link Not Available

Sincerely,

Cheng Gao

Journals Department
Reviewer comments:

Reviewer #1 (Comments for the Author):

Yu et al. investigated the effects of waterlogging and soil type on soybean rhizosphere bacterial structure using three strategies of partial (V4) 16S rRNA sequencing and full-length sequencing of LoopSeq and PacBio. Their results showed that waterlogging significantly influenced the bacterial community structure, with increased relative abundance of *Geobacter*. However, they also stated that whether waterlogging increased the relative abundance of the nitrogen or phosphorus relevant microbes, were dependent on the sequencing methods used according to the co-occurrence network analysis. Be surely, treatment would result in different microbial enrichment which are critical for the stability of bacterial community structure and ecological functions. Altogether, the authors believed that their findings highlight the microbial roles in assisting soybean to resist stress under waterlogging condition, and they confirmed that LoopSeq method could improve the accuracy at species-level environmental samples, and was also cheaper than other two sequencing techniques. Overall, I feel that this study is well organized and has proposed bright viewpoints; the data analyses seem to be solid and the manuscript is well written. I have two suggestions: 1)

please give more details about the sequence processing which would make the reader clearer about what happen (e.g. reads number changes) during each step, these new information would be added in the MM or Results sections, and 2) I was thinking that 16S sequencing would get some of archaeal sequences which were removed or not in this study, if not, mentioning 'bacterial' would be improper, alternatively, using 'prokaryotic' may be more appropriate.

Reviewer #2 (Comments for the Author):

SUMMARY

In this manuscript, the authors test the effects of waterlogging on rhizosphere bacterial communities of two different types of soybean plants (resistant or susceptible to waterlogging), in two different soils (acidic or neutral pH), using three different DNA amplification/sequencing methods (16S V4 PCR/Illumina, 16S full-length PacBio, and 16S full-length LoopSeq).

They had three hypotheses: 1a) waterlogging will decrease bacterial diversity in the soybean rhizosphere and 1b) the effect would be stronger in the acid soil than the neutral soil; 2) taxonomic resolution of microbial communities would be similar in both full-length sequencing technologies, and lower in V4 sequencing; 3) specific bacteria would be enriched in the rhizosphere that may help soybeans resist waterlogging stress.

Addressing these hypotheses, the authors show that:

1. waterlogging had no significant ($p < 0.05$ one-way ANOVA) effect on alpha-diversity except for a lower value (Chao1) in acidic soil with Loopseq.
2. Taxonomic resolution of sequenced communities were distinct in each of three amplification/sequencing technologies when compared to the same database, with slightly more (~97% vs ~95.5%) V4/Illumina sequences assigned to phylum than either LoopSeq or PacBio, but far less assigned to genus (~45% vs. ~88-96%) or species (~15% vs. 30-95%). All assigned proportions are distinct from each other when every sample is regarded as independent ($n=48$).
3. Some bacterial OTUs found in both neutral and acidic soils (60 with V4, 10 with LoopSeq, 9 with PacBio) were relatively more abundant (differential expression analysis) under waterlogging than in control soils.

The authors further show that microbial beta-diversity (CPCoA) was different in each soil and each treatment, but with distinct results dependent on amplification/sequencing technology. Furthermore, that the identities of organisms were drastically different between amplification/sequencing technologies, with the starkest examples being in phylum Acidobacteria (~1/3 as abundant from PacBio compared to either V4/Illumina or LoopSeq) and Planctomycetes (4-5x more abundant in PacBio compared to either V4/Illumina or LoopSeq). The authors discuss the fact that choice of amplification/sequencing method alters the results and interpretation of their waterlogging test; some potentially helpful microbes only appear with differential abundance using some methods. Others (*Geobacter*) appear with all methods. The authors use changes of relative abundance of members of specific modules determined with network analysis to further explore whether waterlogging increased specific microbes that may contribute to helping soybeans resist flooding stress. Without identifying the mechanisms by which these specific microbes do protect soy plants, or even that they do, the authors recognize that though it is reasonable to hypothesize that these microbes may help protect soy plants, this study does not provide evidence that they actually do. The authors conclude with a statement that LoopSeq is a cheaper and higher-utility option than PacBio for full-length 16S sequencing.

REVIEW

I commend the authors for their excellent work! I believe the data and analyses presented in this manuscript would be of immediate and substantial use to many microbial ecologists. However, there are critical issues that must be addressed, mostly omissions of methods and some expected results. These are presented in detail below. In addition, I have a broad suggestion for a re-framing of the data that I would ask the authors consider, but this is only my opinion, and not required for publication.

I have split my comments into three sections: 1) major issues that require amendment, 2) minor edits, and 3) a broad (and totally optional) suggestion for substantial change to the framing of the data and conclusions. The final suggested change would only be to frame the data already presented in this manuscript in a way that I believe may help a wider audience of microbial ecologists engage with it; it is not required in any way.

1. Major Amendments

Line 109 hypotheses ii and iii are well-developed by the previous paragraphs in the introduction, but so far no information has been presented to suggest why waterlogging would decrease bacterial diversity in the soybean rhizosphere, nor why this should be more severe in acidic soil. Either support the first hypothesis in the introduction, or omit this hypothesis.

Line 117 more information is required regarding the soil sampling and preparation. First, soil classification, as detailed as possible, according to any system of soil taxonomy (preferably World Reference Base) is needed for each soil. Second, the approximate sampling depth in cm, plus the number and size of soil samples taken, is needed. Together these are needed to consider the native habitats of the soil microbes in this study. If the soil were taken from the top 5-10 cm with a strong structure,

it is possible that these microbes are never waterlogged for long, and are relatively unadapted to flooding. If the soil were taken from deeper horizons (30+ cm) it is possible that the microbes are relatively well-adapted to hypoxia and waterlogging. In the latter case, it may be the non-waterlogged soils that represent the more intense disturbance from the native habitat. Additionally, it is later stated that the soils were sieved and air-dried but this procedure should be stated here. The process of sampling, sieving, and drying soil up to when it was put in pots should be detailed to the extent that it could be replicated.

Line 118 more information is required about the fate of these two soybean varieties. Despite the fact that two soybean varieties were used, they are never mentioned again throughout the manuscript, though their microbiomes were analyzed. Whether or not these two varieties' microbial communities differed, they should be mentioned again in the results. If they did differ significantly and this is relevant to the conclusions presented in this study, they should be presented and discussed. If they differed significantly but this did not affect the conclusions presented in this study, that should be stated in the results, with the reasoning for not considering soybean variety as an experimental factor explained in the text. If they did not differ significantly, that too should be stated in the results, with that rationale given for why they were treated only as a blocking factor in the statistics (if they were - many tests are given as $n=6$ so I assume they were not treated as independent soy replicates).

Line 122 more detail and structure is needed in this paragraph to get an accurate sense of the experimental design. From what I understand from this paragraph, 6 pots were prepared for each of two soil types, soybean types, and water treatments ($6 \times 2 \times 2 = 48$ pots). 24 pots were filled with neutral soil, 24 pots were filled with acid soil. Eight seeds of either one or the other soybean type were added to each pot. After 6 days past emergence, the number of plants in each pot was thinned down to three. At the V2 growth phase, water was added to the pots until standing 4-6 cm above the soil for waterlogging treatment, while no additional water was added to the control treatment. If this is correct, then all the information is here but the paragraph should be written in a way that would allow for replication: 1) state the number of replicates, experimental factors, and total pots together in the second sentence 2) state how the three remaining soybean plants were selected to remain, while others were removed (Healthiest? Furthest spaced? Random?) 3) State why waterlogging stress examination was performed and what was done with that information 4) state when the water was added (at the V2 stage, right after stress examination?) 5) state how long waterlogging treatment persisted; that is, how long the soils remain saturated. In the following paragraph, it is stated that rhizosphere soil was collected after three days of waterlogging treatment - does that represent three days with 4-6 cm standing water in the pots for all three days? If the soils did drain quickly or it is impossible to know how long the soils were saturated, mention this here.

Line 135 more detail is required regarding soil sampling. How was the rhizosphere soil collected? There are different definitions of what qualifies as 'rhizosphere'. It is implied here that the entire root network of all three soybean plants in each pot were combined in one 50 mL tube for each pot, the roots in the tube was centrifuged, and 5g of soil collected from the bottom of the tube was collected for DNA extraction. If that is true, state the order in which these things were done with sufficient detail (for example how rhizosphere soil was designated, centrifugation speed) such that another lab could replicate it.

Line 142 more detail is required regarding soil analyses. How long was rhizosphere soil stored at 4C prior to analysis? Was rhizosphere soil dried and sieved prior to analysis? Either describe pH measurements and preparation and extraction of soil solution for TN, TK, SOC, NH_4 , NO_3 , TP, and Olsen-P in sufficient detail to replicate these analyses, or cite extraction methods here. Additionally, these data are never reported; please include a table of mean soil data values +/- standard errors for each treatment group in the supplementary material.

Line 177 Unless DADA2 is implemented very differently in QIIME2 than other platforms, DADA2 algorithm produces ASVs to stand in for bacterial organisms, which are distinct from OTUs. This is significant as the difference between ASVs and classified OTUs could account for some of the difference in taxonomy between these amplification/sequencing technologies, especially between LoopSeq (ASVs) and PacBio (OTUs) despite the fact that both are long-read technologies. Clarify in the text when organisms are represented by ASVs vs. OTUs.

Line 198, 298, 432 The authors perform network analysis on microbial communities to analyze co-occurrence between microbes. The authors cite Faust and Raes 2012 to introduce and interpret their network analysis results, but Faust later published a cautionary manuscript demonstrating that the assumptions under which network analysis works are often violated, and the possible inferences overstated, in soil microbial ecology ('From hairballs to hypotheses-biological insights from microbial networks', Rottjers and Faust, 2018). Furthermore, choice of network analysis method and network size can affect network accuracy and meaningfulness, even when assumptions are met ('Difficulty in inferring microbial community structure based on co-occurrence network approaches', Hirano and Takemoto, 2019). The authors must consider at least Rottjers and Faust, 2018 and address the limitations of network analysis within the context of this study, in the discussion section. This is needed to allow readers to fully evaluate the results presented.

Line 216 More information is required about the sequencing results. What sequencing depth was obtained for your samples in each of three sequencing methods (minimum, maximum, and mean)? Were sequenced communities rarefied? If so, how; if not, how was difference in sequencing depth corrected for in diversity analyses?

Line 218 There must be more coherence between statistics as presented in the methods section, results section, the figure captions, and results shown in the figures. For instance, it is stated in the methods section (line 189) that a three-way ANOVA

was performed to assess the difference in bacterial alpha-diversity. It is stated in the results (line 218) that waterlogging had no significant impact on bacterial alpha-diversity (except for acidic soil sequenced by loopseq) - one-way ANOVA, $n=6$, $p > 0.05$. In the caption for this figure (line 830) it states "Effects of waterlogging and soil types on soybean rhizosphere... one-way ANOVA, $n=6$, $p < 0.05$). In the figure itself, letters denoting significant differences correspond to none of these three apparently distinct tests, instead showing the results of all pairwise comparisons, which are never explained in the text. The authors must describe all post-hoc pairwise comparison tests they use in the methods section, with multiple-comparison correction factors, and should discuss significant interaction terms in a previously-conducted 3-way ANOVA rather than calculate a new post-hoc 1-factor ANOVA for each individual comparison of interest.

2. Minor Edits

Line 32 replace "cycle" with "cycling"

Line 34 replace "enriched" with "enriched members"

Line 87 replace "biasness" with "bias", throughout the manuscript

Line 230 the captions on Fig. S1 and S3 are switched.

Line 244+ only genus and species are italicized, all other taxonomic ranks are not.

Line 311 please reference figure 5 again when discussing the volcano plots here. Also - these volcano plots are interesting and important within this manuscript but are almost too small to be interpreted - especially if one were reading the manuscript in print, one would have to take the authors' word for the number and position of ASVs and OTUs involved, since they are too small to be read. I'm not sure if these could be expanded in a separate figure, but the current arrangement is not ideal.

Line 498 please review the references, multiple papers are cited separately multiple times and there are some typos in the paper titles.

Line 826 please define Ne, Ac, W, and CK in this caption. Additionally, consider defining 'CK' in the text - W for Water is self-explanatory but I'm not sure what CK stands for.

Line 851 state how the read abundances were calculated for this figure - averaged across every sample?

Finally, there are many very minor errors in grammar and syntax which don't interfere with the meaning of the text at all, but the authors may wish to get a copy-editor to check it over as a matter of style.

3. Framing - SUGGESTION ONLY, NOT REQUIRED FOR PUBLICATION

The manuscript as written puts first emphasis on the ecological experiment conducted: the effects of water saturation on soy rhizosphere communities in two distinct soils. Throughout the manuscript, the importance of understanding the effects of waterlogging on rhizosphere microbial communities is evaluated before the consideration of amplification and sequencing technology. There is a deep literature on the effects of saturation on soil microbial communities and traits, as the authors cite in their discussion (Evans and Wallenstein 2012 etc). It seems that the authors include the comparison of the three methods (V4-Illumina, most commonly used by microbial ecologists; PacBio, commonly recognized as superior for its longer read length but uncommonly used because of its cost and historically high error rate; and LoopSeq, uncommonly used because of its novelty) as a secondary component of the manuscript. Though the authors' experimental results are certainly interesting and useful to those studying soil microbial adaptations to moisture stress, the exploration of the ways in which selection of a DNA amplification/sequencing technology alters their results is applicable to all microbial ecologists evaluating their own or others' V4 Illumina data, or considering long-read sequencing for future projects.

As the authors say, the field of microbial ecology is significantly based on Illumina V4 sequencing. Though there are many issues with the technology, it is widespread because there is a consensus among microbial ecologists that it is 'good enough' for community profiling. The question I asked myself as I read this manuscript is: would selection of the 'wrong' sequencing technology lead to incorrect results even from a well-constructed study? The authors provide an excellent test of this question, showing exactly how different methods produce slightly different answers to an important question in microbial ecology, though they do not explicitly frame it this way. Instead, the authors tend to focus first on the results of their experiment across all sequencing methods, then only secondly discuss ways in which the method changed the results, if they are mentioned in the text. Even when the differences between sequencing methods are directly addressed, they are often presented as being three equally-valid ways of looking at the community, rather than three different methods, some of which work better than others, that all purport to observe one 'true' community. I believe different results derived from these methods should be cause for interest, analysis, and alarm among microbial ecologists - they cannot all be correct.

The following are four specific examples of what I perceive as the authors' emphasis on the ecological side of these results over

the methodological side, just to give a better sense of what I mean:

1. In the introduction, the authors discuss the issue of crop damage from waterlogging and the possibility that specific microbes may be able to reduce crop damage from waterlogging first (paragraphs 1 and 2) before discussing the issues of lower resolution and inaccurate identification of microbial communities in short-read compared to long read sequencing (paragraphs 3 and 4). However, the authors can and do definitively answer the questions raised by comparing sequencing methods, while they can't address the question of whether any of the microbes they saw actually protect soybeans against waterlogging.

2. In figure 1A, the authors show that both long-read sequencing methods produced communities with Chao1 alpha-diversity ~250, which did not differ by soil type, compared with the short-read sequencing method Chao1 alpha-diversity results of 500-600, which showed significantly more alpha-diversity in neutral soils than acidic soils. The authors have already established that shorter reads lead to less accurate classification of OTUs (lines 88-91); a reasonable conclusion might be that had a researcher relied on V4, they would have come to the (likely false) conclusion that soil type significantly influenced bacterial alpha-diversity, whereas with better sequencing methods (either PacBio or LoopSeq), they would have accurately found that soil type made no (or context-dependent) difference. Instead, the authors only report that there was no overall negative effect of waterlogging on soybean microbe diversity, contrary to their first hypothesis.

3. The effect of amplification/sequencing method on microbial beta-diversity, though seemingly extreme, is not directly compared in Figure 1, which focuses on the effects of soil type and waterlogging instead (Fig. 1C-E, figure S3). How distinct are these communities between sequencing methods, if you plot them on the same CPCoA? And interestingly, when it comes to beta-diversity, the long-read methods do not produce very similar answers (as they did for alpha-diversity) - for some phyla (Proteobacteria) abundances are broadly similar across all methods, for some (Planctomycetes) PacBio yields very different results than either V4 Illumina or LoopSeq, and for some (Acidobacteria) all three methods are distinct. The effect of waterlogging on beta-diversity is mentioned promptly in the discussion (line 356), but the effect of sequencing method on beta-diversity is not mentioned, and the reader has to find it in the supplementary materials.

4. In the discussion, the authors bring up the question of whether they would have observed different results had they used one sequencing method or another (line 381) - but imply by omission throughout the discussion that all three methods are equally valid; that is, that the significant difference in *Variovax* detected in V4 sequencing is just as accurate and useful a finding as the increase in *Pirellula* detected with PacBio. Earlier in this manuscript the authors show that genus-level resolution is far higher in LoopSeq and PacBio than in V4 Illumina, and that species-level resolution is much higher than both in PacBio; why then are results from these three sequencing methods all presented as approximately equally valid and useful?

There are other manuscripts in microbial ecology that use both short-read and long-read sequencing with LoopSeq (<https://doi.org/10.3389/fmicb.2021.598180>) but none that I can find has demonstrated in a way that this manuscript does, how the conclusions of a study might change depending on these sequencing methods used - not just the accuracy of the methods by themselves in reconstructing a mock community, but the way in which the actual results of an experiment do or don't change between methods (for instance, *Geobacter* was detected across methods, other microbes were not). In my opinion, the comparison between amplification/sequencing methods might be due further explicit discussion within the manuscript. Having said this, all these conclusions can be made from a close reading of the data already presented in the manuscript, and the authors do show and explain these differences within the figures and discussion already. The authors should feel free to ignore the comments and suggestions in this section if they disagree with my interpretation.

Staff Comments:

Preparing Revision Guidelines

Please return the manuscript within 60 days; if you cannot complete the modification within this time period, please contact me. If you do not wish to modify the manuscript and prefer to submit it to another journal, please notify me of your decision immediately so that the manuscript may be formally withdrawn from consideration by Microbiology Spectrum.

**Effects of waterlogging on soybean rhizosphere bacterial community using V4, LoopSeq and**
**PacBio 16S rRNA sequence**

Taobing Yu^{a,b}, Lang Cheng^{a,b}, Qi Liu^{a,b}, Shasha Wang^{a,b}, Yuan Zhou^c, Hongbin Zhong^c, Meifang Tang^c,
Hai Nian^{a,b,*}, Tengxiang Lian^{a,b,*}

*^aThe State Key Laboratory for Conservation and Utilization of Subtropical Agro-bioresources, South*
*China Agricultural University, Guangzhou 510642, Guangdong, People's Republic of China*

*^bThe Key Laboratory of Plant Molecular Breeding of Guangdong Province, College of Agriculture,*
*South China Agricultural University, Guangzhou 510642, Guangdong, People's Republic of China*

*^cBGI Genomics, BGI-Shenzhen, Shenzhen 518083, China*

*** Corresponding author1:** Tengxiang Lian

**Corresponding address:** No.483 Wushan Road, Guangzhou, Guangdong, 510642, China.

**Tel:** +86 02085288024

**Fax:** +86 02085288024

**E-mail address:** liantx@scau.edu.cn

*** Corresponding author2:** Hai Nian

**Corresponding address:** No.483 Wushan Road, Guangzhou, Guangdong, 510642, China.

**Tel:** +86 02085288024

**Fax:** +86 02085288024

**E-mail address:** hnian@scau.edu.cn

[revised manuscript text omitted]

that may help soybeans resist to waterlogging stress.

**2. Methods and material**

2.1. Soil and soybean material

A neutral and an acid soil were collected from Yingde County (113°40'N, 24°18'E), and Suixi
County (110°25'N, 21°32'E), Guangdong Province, China, respectively. Two soybean (*Glycine max L.*)
varieties with different waterlogging tolerance, i.e., Qihuang34 (tolerant) and Jidou17 (sensitive), were
used in this study (31, 32).

2.2. Experimental design

We conducted a random pot experiment in the greenhouse at the Agricultural College of South
China Agricultural University, Guangzhou, China. Each treatment on each cultivar was done in six
repetitions. The air-dried soil was sifted through a 2 mm sieve to remove impurities before planting the
soybeans. Each pot (top diameter 13.8 cm, bottom diameter 10.4 cm, height 12.2 cm) used in this study
contained about 2.5 kg of air-dried soil. Eight strong and full soybean seeds with similar shapes were
sown in each pot. The soybean growth process was carried out in a greenhouse with controllable
conditions (temperatures of 26-32 °C in the daytime and 15-21 °C in the nighttime). After 6 days of
emergence, the seedlings were removed from the 3 soybean plants. Waterlogging stress examination
was performed at the soybean in the V2 stage. The water was added to the pots up to 4 to 6 cm above
the soil for the experimental cultivars, while the control plants were left in an ideal environment.

2.3. Soil sampling

After three days of waterlogging treatment, the rhizosphere soil of soybean was collected and the
roots transferred into a 50 ml centrifuge tube filled with phosphate-buffered saline (PBS). After
centrifugation for 10 minutes, 5 g of deposited rhizosphere soil was collected from each sample and
placed in a sterilization centrifuge tube for storage at -80 °C for DNA extraction. The remaining
rhizosphere soil was stored at 4°C prior to the determination of soil physical and chemical properties.

2.4. Analysis of soil properties

Soil pH was determined using a pH meter (FE20-FiveEasy™ pH, Mettler Toledo, Germany) in
soil water suspension (5:1 water-to-soil). Total nitrogen (TN) was determined using an Ultraviolet
Spectrophotometer (UV-1800, Suzhou, China). Total potassium (TK) in soil was measured by flame
atomic absorption spectrometer (AA-7000, Shimadzu, Japan). Soil organic carbon (SOC) content was
assessed using a TOC-5000A analyzer (Shimadzu, Kyoto, Japan). The content of NH₄⁺, NO₃⁻, TP, and
Olsen-P in soil was determined by a continuous flow analytical system (SKALAR SAN⁺⁺,
Netherlands).

**2.5. DNA extraction from soil samples and sequencing process**

Total soil DNA was extracted using a Fast DNA SPIN Kit for Soil (MP Biomedicals, Santa Ana,
CA) following the manufacturer's recommendations. The DNA was eluted with 80 µL H₂O, and
analyzed by Nanodrop 2000 spectrophotometry. Primers 515F (5'-GTGCCAGCMGCCGCGGTAA-3')
and 806R (5'-GGACTACHVGGGTCTAAT-3') with variable 12 bp barcode sequences were used to
amplify the V4 region of the 16S rRNA gene (33). Primers 27F
(5'-AGRGTTYGATYMTGGCTCAG-3') and 1492R (5'-RGYTACCTTGTTACGACTT-3') were used
for the full-length (V1-V9) 16S rRNA gene amplification (LoopSeq and PacBio sequence) (34). The
qPCR reaction system included 22.5 µL PCR SuperMix, 1.0 µL positive primer and 1.0 µL reverse
primer, 10 ng template DNA, dd H₂O supplement to 25 µL. The amplification program was 1 cycle of
95 °C for 60 s, 28 cycles of 95 °C for 60 s, annealing at 58 °C for 60 s and primer extension at 72 °C
for 2 min, and finally 1 cycle of 72 °C for 10 min. Both V4 and full-length sequencing were performed
according to HUADA's standard procedures. Illumina Miseq platform was used to sequence the V4
amplicon (reagent kit v.3; Illumina). The full-length sequencing of PacBio was completed on the
Pacbio RS II platform.

Loopseq-16s-microorganism-24-plex-kit (Loop Genomics, San Jose, CA, USA) was used to
analyze the microbial genome of rhizosphere soil. The unique molecular markers of a single 16S gene
used in LoopSeq sequencing were distributed in the whole gene, and then the full-length 16S gene was
recombined through short reading and sequencing on the Illumina platform. Briefly, 10ng DNA from
different rhizosphere soil samples was used to build a sequencing library. The raw data were collected

by Illumina's NextSeq, with generated FASTQ file (35). All the raw data of the V4 16S sequence,
LoopSeq sequencing, and PacBio sequencing were deposited in the NMDC under accession number
NMDC10017771, NMDC10017785, and NMDC10017787, respectively.

**2.6. Data analysis**

For the V4 16S sequence and LoopSeq, the raw FASTQ sequence file was processed by QIIME 2.
In brief, the Divisive Amplicon Denoising Algorithm 2 (DADA2) in QIIME 2 plugin was used to
obtain OTUs, which detected and corrected amplicon errors and filtered out the potential base error and
chimeric sequences (30, 36). All the raw sequences were filtered, trimmed, and dereplicated. The
representative sequence generated after denoising was based on sklearn's Naive Bayes classifier for
bacterial classification on the SILVA 16S full-length database (36, 37). For the PacBio 16S sequence,
the raw sequence files were processed using SMRT Link software version 5.1.0.26412 (Pacific
Biosciences). The OTUs were clustered using the UPARSE algorithm (38), and parameters were used
to tune the full-length sequencing. The OTUs were iteratively classified according to the latest
non-redundant small subunit SILVA using the RDP-classifier at 97% cutoff.

**2.7. Statistics analyses**

Using the "vegan" package in R V3.6.3, constrained principal-coordinate analysis (CPCoA) based
on the UniFrac distances, adonis test (PERMANOVA), and the mantel test were performed (39).
Genstat V13 was used to perform a three-way analysis of variance (ANOVA) to identify significant

[revised manuscript text omitted]

Contribution

Y.T., L.T., and N.H. conceived and designed the experiments. Y.T., L.T., C.L, L.Q, and W.S. performed
the experiments. Z.Y, Z.H., and T.M. completed the library preparation and sequencing. Y.T., L.T., and
495 N.H. analyzed the data and wrote the paper.

Declaration of conflictions

We declare no conflicts of interest.

References

- 1. Boyer JS, Byrne P, Cassmand KG, Cooper M, Delmer D, Greene T, Gruis F, Habben J,
Hausmann N, Kenny N, Lafitte R, Paszkiewicz S, Porter D, Schlegel A, Schussler J, Setter T,
Shanahan J, Sharp RE, Vyn TJ, Warner D, Gaffney J. 2013. The U.S. drought of 2012 in
perspective: A call to action. *Glob Food Secur-Agr* 2(3): 139–143.
<https://doi.org/10.1016/j.gfs.2013.08.002>.
- 2. Lesk C, Rowhani P, Ramankutty N. 2016. Influence of extreme weather disasters on global crop
production. *Nature* 529(7584): 84–87. <https://doi.org/10.1038/nature16467>.
- 3. Dennis ES, Dolferus R, Ellis M, Muhammad HR, Yongrui W, Hoeren FU, Anil G, Kathleen PI,
Allen GG, Peacock WJ. 2000. Molecular strategies for improving waterlogging tolerance in plants.
*J Exp Bot* 51: 79–82. <https://doi.org/10.1093/jexbot/51.342.89>.
- 4. Kreuzwieser J, Rennenberg H. 2014. Molecular and physiological responses of trees to
waterlogging stress. *Plant Cell Environ* 37(10): 2245–2259. <https://doi.org/10.1111/pce.12310>.
- 5. Sairam RK, Kumutha D, Ezhilmathi K, Chinnusamy V, Meena RC. 2009. Waterlogging induced
oxidative stress and antioxidant enzyme activities in pigeon pea. *Biol Plantarum* 53: 493–504.
<https://doi.org/10.1007/s10535-009-0090-3>.
- 6. Grichko VP, Glick BR. 2001. Amelioration of flooding stress by ACC deaminasecontaining plant
growth-promoting bacteria. *Plant Physiol Bioch.* 39(1): 11–17.
[https://doi.org/10.1016/S0981-9428\(00\)01212-2](https://doi.org/10.1016/S0981-9428(00)01212-2).
- 7. Yang H, Sheng R, Zhang Z, Ling W, Qing W, Wei WX. 2016. Responses of nitrifying and
denitrifying bacteria to flooding drying cycles in flooded rice soil. *Appl Soil Ecol* 103: 101–109.
<https://doi.org/10.1016/j.apsoil.2016.03.008>.
- 8. Neatrou MA, Webster JR, Benfield EF. 2004. The role of floods in particulate organic matter
dynamics of a southern Appalachian river-floodplain ecosystem. *J North Am Benthol Soc.* 23(2):
198–213. [https://doi.org/10.1899/0887-3593\(2004\)023<0198:TROFIP>2.0.CO;2](https://doi.org/10.1899/0887-3593(2004)023<0198:TROFIP>2.0.CO;2).
- 9. Sánchez-Rodríguez AR, Hill PW, Chadwick DR, Jones DL. 2019. Typology of extreme flood
event leads to differential impacts on soil functioning. *Soil Biol Biochem* 129: 153–168.
<https://doi.org/10.1016/j.soilbio.2018.11.019>.

- 10. Ngumbi E, Kloepper J. 2016. Bacterial-mediated drought tolerance: Current and future prospects.
*Appl Soil Ecol* 105: 109 – 125. <https://doi.org/10.1016/j.apsoil.2016.04.009>.
- 11. Kang SM, Khan AL, Waqas M, Young HY, Jin HK, Jong GK, Muhammad H, In JL. 2014. Plant
growth-promoting rhizobacteria reduce adverse effects of salinity and osmotic stress by regulating
phytohormones and antioxidants in *Cucumis sativus*. *J Plant Interact* 9: 673–682.
<https://doi.org/10.1080/17429145.2014.894587>.
- 12. Nascimento FX, Rossi MJ, Soares CR, Brendan JM, Bernard RG. 2014. New insights into
1-aminocyclopropane-1-carboxylate (ACC) deaminase phylogeny evolution and ecological
significance. *PLoS One* 9(6): e99168. <https://doi.org/10.1371/journal.pone.0099168>.
- 13. Armada E, Azcon R, Lopez C, Calvo PM, Ruiz-Lozano JM. 2015. Autochthonous arbuscular
mycorrhizal fungi and *Bacillus thuringiensis* from a degraded Mediterranean area can be used to
improve physiological traits and performance of a plant of agronomic interest under drought
conditions. *Plant Physiol Bioch.* 90: 64–74. <https://doi.org/10.1016/j.plaphy.2015.03.004>.
- 14. Lakshmanan V, Ray P, Craven KD. 2017. Toward a resilient functional microbiome: drought
tolerance-alleviating microbes for sustainable agriculture. *Methods Mol Biol* 1631: 69–84.
https://doi.org/10.1007/978-1-4939-7136-7_4.
- 15. Mentzer JL, Goodman RM, Balsler TC. 2006. Microbial response over time to hydrologic and
fertilization treatments in a simulated wet prairie. *Plant Soil* 284: 85–100.
<https://doi.org/10.1007/s11104-006-0032-1>.
- 16. Suzuki C, Kunito T, Aono T, Liu CT, Oyaizu H. 2005. Microbial indices of soil fertility. *J Appl*
*Microbiol* 98: 1062 – 1074. <https://doi.org/10.1111/j.1365-2672.2004.02529.x>.
- 17. Evans SE, Wallenstein MD. 2012. Soil microbial community response to drying and rewetting
stress: does historical precipitation regime matter. *Biogeochemistry* 109: 101–116.
<https://doi.org/10.2307/41490547>.
- 18. Kozich JJ, Westcott SL, Baxter NT, Sarah KH, Patrick DS. 2013. Development of a dual-index
sequencing strategy and curation pipeline for analyzing amplicon sequence data on the miseq
illumina sequencing platform. *Appl Environ Microb.* 79(17): 5112–5120.
<https://doi.org/10.1128/AEM.01043-13>.
- 19. Schloss PD, Eisen JA. 2010. The effects of alignment quality distance calculation method
sequence filtering and region on the analysis of 16S rRNA gene-based studies. *PLOS Comput*

- Biol 6(7): e1000844. <https://doi.org/10.1371/journal.pcbi.1000844>.
- 20. Youssef N, Sheik CS, Krumholz LR, Fares ZN, Bruce AR, Mostafa SE. 2009. Comparison of
species richness estimates obtained using nearly complete fragments and simulated
pyrosequencing-generated fragments in 16S rRNA gene-based environmental surveys. *Appl*
*Environ Microbiol* 75(16): 5227 – 5236. <https://doi.org/10.1128/AEM.00592-09>.
- 21. Gilbert JA, Meyer F, Antonopoulos D, Pavan B, Brown CT, Christopher TB, Narayan D, Jonathan
AE, Dir KE, Dawn F, Wu F, Daniel H, Janet J, Rob K, James K, Eugene K, Kostas K, Joel K,
Nikos K, Rachel M, Alice M, Christopher Q, Jeroen R, Alexander S, Ashley S, Rick S. 2010.
Meeting report: the terabase metagenomics workshop and the vision of an Earth microbiome
project. *Stand Genomic Sci.* 3(3): 243–248. <https://doi.org/10.4056/sigs.1433550>.
- 22. Huse SM, Dethlefsen L, Huber JA, Welch DM, Relman DA, Sogin ML. 2008. Exploring
microbial diversity and taxonomy using SSU rRNA hypervariable tag sequencing. *PLoS Genet.*
4(11): e1000255. <https://doi.org/10.1371/journal.pgen.1000255>.
- 23. Claesson MJ, O'Sullivan O, Wang Q, Janne N, Julian RM, Hauke S, Willem MV, Ross RP,
O'Toole PW. 2009. Comparative analysis of pyrosequencing and a phylogenetic microarray for
exploring microbial community structures in the human distal intestine. *PLoS One* 4(8): e6669.
<https://doi.org/10.1371/journal.pone.0006669>.
- 24. Esther S, Brian B, Devin CD, Brett B, Robert MB, Asaf L, Esther AG, Jan FC, Alex C, Hans PK,
Steven JH, Philip H, Susannah GT, Tanja W. 2016. High-resolution phylogenetic microbial
community profiling. *ISME J.* 10: 2020–2032. <https://doi.org/10.1038/ismej.2015.249>.
- 25. Tremblay J, Singh K, Fern A, Kirton ES, Tringe SG, Woyke T, Janey L, Feng C, Jeffery LD,
Susannah GT. 2015. Primer and platform effects on 16S rRNA tag sequencing. *Front Microbiol* 6:
771. <https://doi.org/10.3389/fmicb.2015.00771>.
- 26. Guo F, Ju F, Cai L, Zhang T. 2013. Taxonomic precision of different hypervariable regions of 16S
rRNA gene and annotation methods for functional bacterial groups in biological wastewater
treatment. *PLoS One* 8(10): e76185. <https://doi.org/10.1371/journal.pone.0076185>.
- 27. Earl JP, Adappa ND, Krol J, Bhat AS, Balashov S, Ehrlich RL, Palmer JN, Workman AD, Blasetti
583 M, Sen B, Hammond J, Cohen NA, Ehrlich GD, Mel JC. 2015. Species-level bacterial community
profiling of the healthy sinonasal microbiome using Pacific Biosciences sequencing of full-length
16S rRNA genes. *Microbiome* 6(1): 190. <https://doi.org/10.1186/s40168-018-0569-2>.

- 28. Rosalinda D, Umer ZI, Melanie S, John GK, Richard G, Alistair CD, Migun S, Mircea P,
Christophe Q, Neil H. 2016. A comprehensive benchmarking study of protocols and sequencing
methods for 16 S rRNA community profiling. *BMC Genomics* 17: 55.
<https://doi.org/10.1186/s12864-015-2194-9>.
- 29. Callahan BJ, Grinevich D, Thakur S, Balamotis MA, Yehezkel TB. 2021 .Ultra-accurate microbial
amplicon sequencing with synthetic long reads. *Microbiome* 9(1): 1-13.
<https://doi.org/10.1186/s40168-021-01072-3>.
- 30. Jeong J, Yun K, Mun S, Chung WH, Choi SY, Young-do N, Lim MY, Hong CY, Park CH, Ahn YJ,
Han K. 2021. The effect of taxonomic classification by full-length 16S rRNA sequencing with a
synthetic long-rad technology. *Sci Rep-UK* 11: 10861.
<https://doi.org/10.1038/s41598-020-80826-9>.
- 31. Xu R, Wang CJ, Zhang LF, Li W, Dai HY, Zhang J. 2013. Breeding of a new soybean variety
Qihuang 34 with high yield high quality and multi resistance. *Shandong Agricultural Sciences* 45:
107–108.
- 32. Zhao QS, Yan L, Liu BQ, Di R, Shi XL, Zhao SJ, Zhang MC, Yang CY. 2015. Breeding of
High-yield Widespread and High-quality Soybean Cultivar Jidou 17. *Soybean Science* 34: 736 –
739.
- 33. Caporaso JG, Lauber CL, Walters WA, Donna BL, James H, Noah F, Sarah MO, Jason B, Louise
F, Markus B, Niall G, Jack AG, Geoff S, Rob K. 2012. The ISMEUltra-high-throughput microbial
community analysis on the Illumina HiSeq and MiSeq platforms. *ISME J* 6: 1621–1624.
<https://doi.org/10.1038/ismej.2012.8>.
- 34. Moshier JJ, Bernberg EL, Shevchenko O, Kan J, Kaplan LA. 2013. Efficacy of a 3rd generation
high-throughput sequencing platform for analyses of 16S rRNA genes from environmental
samples. *J Microbiol Meth* 95: 175 – 181. <https://doi.org/10.1016/j.mimet.2013.08.009>.
- 35. Wallis KF, Melnyk SB, Miousse IR. 2020. Sex-Specific effects of dietary methionine restriction on
the intestinal microbiome. *Nutrients* 12(3): 781. <https://doi.org/10.3390/nu12030781>.
- 36. Park C, Kyeong EY, Jeong MC, Ji YL, Chang PH, Young DN, Jinuk J, Kyudong H, Yong JA.
2020. Performance comparison of fecal preservative and stock solutions for gut microbiome
storage at room temperature. *J Microbiol* 58: 703–710.
<https://doi.org/10.1007/s12275-020-0092-6>.

- 37. Cole JR, Wang Q, Fish JA, Chai B, Mcgarrell DM, Sun Y, Brown CT, Porras AA, Kuske CR,
Tiedje JM. 2014. Ribosomal Database Project: Data and tools for high throughput rRNA analysis.
*Nucleic Acids Res* 42: 633–642. <https://doi.org/10.1093/nar/gkt1244>.
- 38. Edgar RC. 2013. UPARSE: highly accurate OTU sequences from microbial amplicon reads. *Nat.*
*Methods* 10(10) 996 – 98. <https://doi.org/10.1038/nmeth.2604>.
- 39. Anderson MJ. 2001. A new method for non-parametric multivariate analysis of variance. *Austral*
*Ecol* 26(1): 32 – 46. <https://doi.org/10.1111/j.1442-9993.2001.tb00081.x>.
- 40. Lian T, Wang G, Yu Z, Li YS, Liu XB, Jian J. 2016. Carbon input from ¹³C-labelled soybean
residues in particulate organic carbon fractions in a mollisol. *Biol Fert Soils* 52: 331 – 339.
<https://doi.org/10.1007/s00374-015-1080-6>.
- 41. Edwards J, Johnson C, Santos MC, Eugene L, Natraj KP, Srijak B, Jonathan AE, Venkatesan S.
2015. Structure variation and assembly of the root-associated microbiomes of rice. *PNAS* 112(8):
911–920. <https://doi.org/10.1073/pnas.1414592112>.
- 42. Chong J, Liu P, Zhou G, Xia J. 2020. Using MicrobiomeAnalyst for comprehensive statistical
functional and meta-analysis of microbiome data. *Nat Protoc* 5: 799–821.
<https://doi.org/10.1038/s41596-019-0264-1>.
- 43. Jiang Y, Li S, Li R, Jia Z, Liu YH, Lian FL, Hong Z, Wu WL, Li WL. 2017. Plant cultivars
imprint the rhizosphere bacterial community composition and association networks. *Soil Biol*
*Biochem* 109: 145–155. <https://doi.org/10.1016/j.soilbio.2017.02.010>.
- 44. Agler MT, Jonas R, Samuel K, Constanze M, Sang TK, Detlef W, Eric MK. 2016. Microbial hub
taxa link host and abiotic factors to plant microbiome variation. *PLOS Biol* 14(1): e1002352.
<https://doi.org/10.1371/journal.pbio.1002352>.
- 45. Diniz-Filho JAF, Soares TN, Lima JS, Dobrovolski R, Landeiro VL, Telles MPDC, Rangel TF,
Bini LM, 2013. Mantel test in population genetics. *Genet Mol Biol* 36(4): 475–485.
<https://doi.org/10.1590/S1415-47572013000400002>.
- 46. Evans SE, Wallenstein MD. 2012. Soil microbial community response to drying and rewetting
stress: does historical precipitation regime matter. *Biogeochemistry* 109: 101–116.
<https://doi.org/10.2307/41490547>.
- 47. Irene MU, Ann CK, Rose MM. 2009. Flooding effects on soil microbial communities. *Appl Soil*
*Ecol* 42(1): 1–8. <https://doi.org/10.1016/j.apsoil.2009.01.007>.

- 48. Ponnamperna FN. 1972. The chemistry of submerged soils. *Adv Agron.* 24: 29–96.
[https://doi.org/10.1016/S0065-2113\(08\)60633-1](https://doi.org/10.1016/S0065-2113(08)60633-1).
- 49. Preece C, Peñuelas J. 2016. Rhizodeposition under drought and consequences for soil
communities and ecosystem resilience. *Plant Soil* 409: 1–17.
<https://doi.org/10.1007/s11104-016-3090-z>.
- 50. Meisner A, Leizeaga A, Rousk J, Bååth E. 2017. Partial drying accelerates bacterial growth
recovery to rewetting. *Soil Biol Biochem* 112: 269–276.
<https://doi.org/10.1016/j.soilbio.2017.05.016>.
- 51. Nguyen L, Osanai Y, Anderson IC, Michael PB, Michael B, David TT, Brajesh KS. 2018a.
Impacts of waterlogging on soil nitrification and ammonia-oxidizing communities in farming
system. *Plant Soil* 426: 1–13. <https://doi.org/10.1007/s11104-018-3584-y>.
- 52. Sanaullah M, Blagodatskaya E, Chabbi A, Cornelia R, Yakov K. 2011. Drought effects on
microbial biomass and enzyme activities in the rhizosphere of grasses depend on plant community
composition. *Appl. Soil Ecol.* 48 38 – 44. <https://doi.org/10.1016/j.apsoil.2011.02.004>.
- 53. Canarini A, Dijkstra F. 2015. Dry-rewetting cycles regulate wheat carbon rhizodeposition
stabilization and nitrogen cycling. *Soil Biol Biochem.* 81: 195–203.
<https://doi.org/10.1016/j.soilbio.2014.11.014>.
- 54. Fuchslueger L, Bahn M, Fritz K, Roland H, Andreas R. 2014. Experimental drought reduces the
transfer of recently fixed plant carbon to soil microbes and alters the bacterial community
composition in a mountain meadow. *New Phytol* 201(3): 916–927.
<https://doi.org/10.1111/nph.12569>.
- 55. Woese CR, Magrum LJ, Gupta R, Siegel RB, Stahl DA, Kop J, Crawford N, Brosius J, Gutell R,
Hogan JJ, Noller HF. 1980. Secondary structure model for bacterial 16S ribosomal RNA:
phylogenetic enzymatic and chemical evidence. *Nucleic Acids Res* 8(10): 2275–2293.
<https://doi.org/10.1093/nar/8.10.2275>.
- 56. Fichot EB, Norman SR. 2013. Microbial phylogenetic profiling with the Pacific Biosciences
sequencing platform. *Microbiome* 1: 10. <https://doi.org/10.1186/2049-2618-1-10>.
- 57. Tremblay J, Singh K, Fern A, Kirton ES, Tringe SG, Woyke T, Janey L, Feng C, Jeffery LD,
Susannah GT. 2015. Primer and platform effects on 16S rRNA tag sequencing. *Front Microbiol* 6:
771. <https://doi.org/10.3389/fmicb.2015.00771>.

- 58. Johnson JS, Spakowicz DJ, Hong BY, Weinstock GM. 2019. Evaluation of 16S rRNA gene
sequencing for species and strain-level microbiome analysis. *Nat Commun* 10(1): 5029.
<https://doi.org/10.1038/s41467-019-13036-1>.
- 59. Liu Z, Lozupone C, Hamady M, Frederic DB, Rob K. 2007. Short pyrosequencing reads suffice
for accurate microbial community analysis. *Nucleic Acids Res* 35(18): e120.
<https://doi.org/10.1093/nar/gkm541>.
- 60. Oscar F, Hu JH, Bao XL, Steven HI, Inga P, Ali B. 2015. Improved OTU-picking using long-read
16 S rRNA gene amplicon sequencing and generic hierarchical clustering. *Microbiome* 3: 43.
<https://doi.org/10.1186/s40168-015-0105-6>.
- 61. Johnson JS, Spakowicz DJ, Hong BY, Weinstock GM. 2019. Evaluation of 16S rRNA gene
sequencing for species and strain-level microbiome analysis. *Nat Commun*. 10(1): 5029.
<https://doi.org/10.1038/s41467-019-13036-1>.
- 62. Liu Z, DeSantis TZ, Andersen GL, Knight R. 2008. Accurate taxonomy assignments from 16S
rRNA sequences produced by highly parallel pyrosequencers. *Nucleic Acids Res* 36(18): e120.
<https://doi.org/10.1093/nar/gkn491>.
- 63. Phillip R, Myer MK, Harvey CF, Timothy PLS. 2016. Evaluation of 16S rRNA amplicon
sequencing using two next-generation sequencing technologies for phylogenetic analysis of the
rumen bacterial community in steers. *J Microbiol Meth* 127: 132–140.
<https://doi.org/10.1016/j.mimet.2016.06.004>.
- 64. Omri MF, Isai SG, Gabriel C, Jonathan MC, Theresa FL, Paulo JPLT, Ellie DW, Connor RF,
Corbin DJ, Jeffery LD. 2020. A single bacterial genus maintains root growth in a complex
microbiome. *Nature* 587(7832): 103–108. <https://doi.org/10.1038/s41586-020-2778-7>.
- 65. Wang H, Li Z, Han H. 2017a. Comparison of different ecological remediation methods for
removing nitrate and ammonium in Qinshui River Gonghu Bay Taihu Lake. *Environ Sci Pollut R*
4: 1706–1718. <https://doi.org/10.1007/s11356-016-7963-8>.
- 66. Zhang WW, Wang C, Xue R, Wang LJ. 2019a. Effects of salinity on the soil microbial community
and soil fertility. *J Integr Agr* 18(6): 1360–1368. [https://doi.org/10.1016/s2095-3119\(18\)62077-5](https://doi.org/10.1016/s2095-3119(18)62077-5).
- 67. Kunkun F, Delgado-Baquerizo M, Zhu YG, Chu HY. 2020. Crop production correlates with soil
multitrophic communities at the large spatial scale. *Soil Biol Biochem* 151: 108047.
<https://doi.org/10.1016/j.soilbio.2020.108047>.

- 68. Jia R, Wang K, Li L, Zhi Q, Shen WH, Dong Q. 2020. Abundance and community succession of
nitrogen-fixing bacteria in ferrihydrite enriched cultures of paddy soils is closely related to
Fe(III)-reduction. *Sci Total Environ.* 720: 137633.
<https://doi.org/10.1016/j.scitotenv.2020.137633>.
- 69. Yuan YB, Jan D, Wang BZ, Chen RR, Miansong H, Li ZP, Lin XG, Feng YZ. 2019. Bacterial
communities involved directly or indirectly in the anaerobic degradation of cellulose. *Biol Fert*
*Soils* 55: 201 – 211. <https://doi.org/10.1007/s00374-019-01342-1>.
- 70. Kato S, Hashimoto K, Watanabe K. 2011. Methanogenesis facilitated by electric syntrophy via
(semi) conductive iron-oxide minerals. *Environ Microbiol* 14: 1646–1654.
<https://doi.org/10.1111/j.1462-2920.2011.02611.x>.
- 71. Li HJ, Chang JL, Liu PF, Li F, Dewen D, Lu YH. 2015. Direct interspecies electron transfer
accelerates syntrophic oxidation of butyrate in paddy soil enrichments. *Environ Microbiol* 17(5):
1533–1547. <https://doi.org/10.1111/1462-2920.12576>.
- 72. Jing XY, Yang YT, Ai ZH, Chen SS, Zhou SG. 2019. Potassium channel blocker inhibits the
formation and electroactivity of *Geobacter* biofilm. *Sci Total Environ* 705: 135796.
<https://doi.org/10.1016/j.scitotenv.2019.135796>.
- 73. Nguyen LTT, Osanai Y, Lai K, Ian CA, Michael PB, David TT, Brajesh KS. 2018b. Responses of
the soil microbial community to nitrogen fertilizer regimes and historical exposure to extreme
weather events: Flooding or prolonged-drought. *Soil Biol Biochem* 118: 227–236.
<https://doi.org/10.1016/j.soilbio.2017.12.016>.
- 74. Zhang JY, Yong XL, Na Z, Bin H, Tao J, Hao X, Yuan Q, Yan PX, Zhang XN, Guo XX, Jing H,
Cao SY, Xin W, Chao W, Hui W, Qu BY, Fan GY, Yuan LX, Ruben GO, Chu CC, Yang B. 2019b.
NRT1.1B is associated with root microbiota composition and nitrogen use in field-grown rice. *Nat*
*Biotechnol* 37(6): 676–684. <https://doi.org/10.1038/s41587-019-0104-4>.
- 75. Sarkar A, Kazy SK, Sar P. 2014. Studies on arsenic transforming groundwater bacteria and their
role in arsenic release from subsurface sediment. *Environmental Science and Pollution Research*
21: 8645–8662. <https://doi.org/10.1007/s11356-014-2759-1>.
- 76. Islam FS, Gault G, Boothman C, David AP, John MC, Debashis C. Jonathan RL. 2004. Role of
metal-reducing bacteria in arsenic release from Bengal delta sediments. *Nature* 430(6995): 68–71.
<https://doi.org/68-71.10.1038/nature02638>.

- 77. Lian TX, Wang GH, Yu ZH, Li YS, Liu XB, Zhang SQ, Stephen JH, Jian J. 2017. Bacterial
communities incorporating plantderived carbon in the soybean rhzosphere in Mollisols that differ
in soil organic carbon content. *Appl Soil Ecol* 119: 375–383.
<https://doi.org/10.1016/j.apsoil.2017.07.016>.
- 78. Zhou T, Wang L, Du YL, Liu T, Li SX, Gao Y, Liu WG, Yang WY. 2019. Rhizosphere soil
bacterial community composition in soybean genotypes and feedback to soil P availability. *J*
*Integr Agr* 18(10): 2230 – 2241. [https://doi.org/10.1016/s2095-3119\(18\)62115-x](https://doi.org/10.1016/s2095-3119(18)62115-x).
- 79. Meng FQ, Qiao YH, Wu WL, Pete S, Steffanie S. 2017. Environmental impacts and production
performances of organic agriculture in China:A monetary valuation. *J Environ Manage* 188: 49–
57. <https://doi.org/10.1016/j.jenvman.2016.11.080>.
- 80. Long XE, Yao H, Huang Y, Wei WX, Zhu YG. 2018. Phosphate levels influence the utilisation of
rice rhizodeposition carbon and the phosphate-solubilising microbial community in a paddy soil.
*Soil Biol Biochem* 118: 103–114. [/https://doi.org/10.1016/j.soilbio.2017.12.014](https://doi.org/10.1016/j.soilbio.2017.12.014).
- 81. Wang R, Sun Q, Wang Y, Liu QF, Du LL, Zhao M, Gao X, Hu YX, Guo SL. 2017b. Temperature
sensitivity of soil respiration: Synthetic effects of nitrogen and phosphorus fertilization on chinese
Loess Plateau. *Sci Total Environ* 574: 1665–1673. <https://doi.org/10.1016/j.scitotenv.2016.09.001>.
- 82. Hirokazu T, Kabir GP, Masato Y, Kazuhiko N, Kei H, Ken N, Shinji F, Masayuki U, Shinji N,
Yusuke O, Kentaro Y, Klaus S, Yang B, Ryo S, Yasunori I, Kiwamu M, Kiers ET. 2018. Core
microbiomes for sustainable agroecosystems. *Nat Plants* 4: 247–257.
<https://doi.org/10.1038/s41477-018-0139-4>.
- 83. Xun W, Liu Y, Li W, Yi R, Wu X, Xu ZH, Nan Z, Miao YZ, Shen QR, Zhang RF. 2021.
Specialized metabolic functions of keystone taxa sustain soil microbiome stability. *Microbiome*
9(1): 35. <https://doi.org/10.1186/s40168-020-00985-9>.
- 84. Anne D, Katharina K, Hanna K, Craig WH, Michaela S, Jasmin S, Thomas Z, Søren MK, Mads A,
Per HN, Michael W, Holger D. 2020. Exploring the upper pH limits of nitrite oxidation: diversity
ecophysiology and adaptive traits of haloalkalitolerant Nitrospira. *ISME J* 14: 2967–2979.
<https://doi.org/10.1038/s41396-020-0724-1>.
- 85. Zhong YQW, Hu JH, Xia QM, Zhang SL, Xin L, Pan XY, Zhao RP, Wang RW, Yan WM, Shang
GZP, Hu FY, Yang CD, Wen W. 2020. Soil microbial mechanisms promoting ultrahigh rice yield.
*Soil Biol Biochem* 143: 107741. <https://doi.org/10.1016/j.soilbio.2020.107741>.

- 86. Angana S, Sufia KK, Pinaki S. 2014. Studies on arsenic transforming groundwater bacteria and
their role in arsenic release from subsurface sediment. *Environ. Sci Pollut R* 21: 8645–8662.
<https://doi.org/10.1007/s11356-014-2759-1>.
- 87. Faust K, Raes J. 2012. Microbial interactions: from networks to models. *Nat Rev Microbiol* 10:
538–550. <https://doi.org/10.1038/nrmicro2832>.
- 88. Ziegler M, Eguíluz VM, Duarte CM, Voolstra CR. 2018. Rare symbionts may contribute to the
resilience of coral – algal assemblages. *ISME J* 12(1): 161–72.
<https://doi.org/10.1038/ismej.2017.151>.
- 89. Kara EL, Hanson PC, Hu YH, Luke W, Katherine DM. 2013. A decade of seasonal dynamics and
co-occurrences within freshwater bacterioplankton communities from eutrophic Lake Mendota
WI USA. *ISME J* 7(3): 680–684. <https://doi.org/10.1038/ismej.2012.118>.
- 90. Olesen JM, Bascompte J, Dupont YL, Jordano P. 2007. The modularity of pollination networks.
*PNAS* 104: 19891–19896. <https://doi.org/10.1073/pnas.0706375104>.
- 91. Wu X, Alexandre J, Sai G, Ida K, Zhao QY, Wu HS, George AK, Shen QR, Li R, Stefan G. 2018.
Soil protist communities form a dynamic hub in the soil microbiome. *ISME J* 12: 634–638.
<https://doi.org/10.1038/ismej.2017.171>.
- 92. Chen S, Tatoba RW, Ruibo S, Kuramae EE, Liu B. 2019. Root-associated microbiomes of wheat
under the combined effect of plant development and nitrogen fertilization. *Microbiome* 7(1): 136.
<https://doi.org/10.1186/s40168-019-0750-2>.
- 93. Kunkun F, Delgado-Baquerizo M, Zhu YG, Chu HY. 2020. Crop production correlates with soil
multitrophic communities at the large spatial scale. *Soil Biol Biochem* 151: 108047.
<https://doi.org/10.1016/j.soilbio.2020.108047>.
- 94. Lalucat J, Bennasar A, Bosch R, Elena GV, Norberto JP. 2006. Biology of *Pseudomonas stutzeri*.
*Microbiol. Mol Biol Rev* 70(2): 510–547. <https://doi.org/10.1128/MMBR.00047-05>.
- 95. Patricia CDS, Dennis RD. 2011. Co-ordination and fine-tuning of nitrogen fixation in *Azotobacter*
*vinelandii*. *Mol Microbiol* 79: 1132–1135. <https://doi.org/10.1111/j.13>.
- 96. Baskaran V, Patil PK, Antony ML, Avunje S, Nagaraju VT, Ghate SD, Nathamuni S,
Dineshkumar N, Alavandi SV, Vijayan KK. 2020. Microbial community profiling of ammonia and
nitrite oxidizing bacterial enrichments from brackishwater ecosystems for mitigating nitrogen
species. *Sci Rep-UK* 10: 5201. <https://doi.org/10.1038/s41598-020-62183-9>.

- 97. Liao SL, Wang YY, Liu H, Fan GY, Sunil KS, Tao J, Chen JW, Zhang PF, Lone G, Mikael LS, Shi
Q, Simon M, Yuen L, Liu X. 2020. Deciphering the microbial taxonomy and functionality of two
diverse mangrove ecosystems and their potential abilities to produce bioactive compounds.
*mSystems* 5(5): e00851–19. <https://doi.org/10.1128/mSystems.00851-19>.
- 98. Xun W, Liu Y, Li W, Yi R, Wu X, Xu ZH, Nan Z, Miao YZ, Shen QR, Zhang RF. 2021.
Specialized metabolic functions of keystone taxa sustain soil microbiome stability. *Microbiome*
9(1): 35. <https://doi.org/10.1186/s40168-020-00985-9>.
- 99. Xu J, Liu SJ, Song SR, Guo HL, Tang JJ, Yong JWH, Ma YD, Chen X. 2019. Arbuscular
mycorrhizal fungi influence decomposition and the associated soil microbial community under
different soil phosphorus availability. *Soil Biol. Biochem.* 120: 181–190.
<https://doi.org/10.1016/j.soilbio.2018.02.010>.
- 100. Ma B, Wang H, Dsouza M, Jun L, Yan H, Dai ZM, Philip CB, Xu JM, Jack AG. 2015. Geographic
patterns of co-occurrence network topological features for soil microbiota at continental scale in
eastern China. *ISME J* 10: 1891 – 1901. <https://doi.org/10.1038/ismej.2015.261>.
- 101. Fiona MS, Paul BLG, Inma L, Davey LJ, Simon C, David AR. 2020. Soil textural heterogeneity
impacts bacterial but not fungal diversity. *Soil Biol Biochem* 144: 107766.
<https://doi.org/10.1016/j.soilbio.2020.107766>.
- 102. Trivedi P, Leach JE, Tringe SG, Sa T, Singh BK. 2021. Plant - microbiome interactions: from
community assembly to plant health. *Nat Rev Microbiol* 18(11): 607–621.
<https://doi.org/10.1038/s41579-020-0412-1>.
- 103. Tao K, Kelly S, Radutoiu S. 2019. Microbial associations enabling nitrogen acquisition in plants.
*Curr Opin Microbiol* 49: 83–89. <https://doi.org/10.1016/j.mib.2019.10.005>.
- 104. Yu XJ. Chen Q. Shi WC. Gao Z. Sun X. Dong JJ. Li J. Wang HT. Gao JG. Liu ZG. Zhang M.
2021. Interactions between phosphorus availability and microbes in a wheat-maize double
cropping system: a reduced fertilization scheme. *J Integr Agr* 20(0): 2–16.
[https://doi.org/10.1016/S2095-3119\(20\)63599-7](https://doi.org/10.1016/S2095-3119(20)63599-7).

Table 1 Related attributes of different sequencing platforms

	V4	LoopSeq	PacBio
Cloning required	No	No	No
Average sequence time	8h	2.5h	2 h/SMRT cell
Average read length	~250bp	~1500bp	1000-1500bp
Read technology	Short-read technology	Long-reads technology	Long-reads technology
Sequencing variable region	V4	V1-V9	V1-V9
Stitching during sequencing	Yes	No	No
Error rates	high	lower	medium
Eliminate PCR bias	NO	Yes	Yes
Need stitching	Yes	NO	NO
Approximate cost per Mb	US\$0.11	US\$0.245	US\$2.50

824

825

826 Table 2 Effects of waterlogging and soil types on bacterial community structure in soybean
 827 rhizosphere analyzed by permutational multivariate analysis of variance (PERMANOVA).

Sequencing methods	Factor	F	R ²	P
V ₄	Soil	111.7140	0.66882	0.001 ***
	Water	10.2152	0.06116	0.002 **
	Soil:Water	1.1018	0.00660	0.246
	NeW Vs NeCK	7.0589	0.24292	0.001 ***
	AcW Vs AcCK	6.7174	0.23391	0.001 ***
LooSeq	Soil	17.5165	0.26413	0.001 ***
	Water	2.6542	0.04002	0.012 *
	Soil:Water	2.1465	0.03237	0.026 *
	NeW Vs NeCK	2.5809	0.105	0.002 **
	AcW Vs AcCK	3.0537	0.12189	0.001 ***
PacBio	Soil	0.9707	0.01801	0.358
	Water	7.9983	0.14838	0.001 ***
	Soil:Water	0.9332	0.01731	0.386
	NeW Vs NeCK	2.5963	0.10556	0.001 ***
	AcW Vs AcCK	7.9223	0.26476	0.001 ***

* indicates significant value of $P < 0.05$, ** indicates significant value of $P < 0.01$, *** indicates
 significant value of $P < 0.001$.

Fig. 1. Effects of waterlogging and soil types on soybean rhizosphere soil bacterial (A) Chao1 and (B)
Shannon index with Illumina Miseq, LoopSeq, and PacBio full-length sequencing method (one-way
ANOVA, $n=6$, $P < 0.05$). (CDE) Constrained Principal-coordinate analysis (CPCoA) based on
Bray-Curtis distance showing differences in rhizosphere bacterial community structure under
waterlogging in neutral and acidic soil (PERMANOVA, $n=6$, $P < 0.05$). Different letters indicate
significant differences ($P < 0.05$). NeCK: Soybean rhizosphere soil without waterlogging in neutral
soil; NeW: Soybean rhizosphere soil with waterlogging in neutral soil; AcCK: Soybean rhizosphere
soil without waterlogging in acidic soil; AcW: Soybean rhizosphere soil with waterlogging in acid soil.
V4: Illumina Miseq; LoopSeq: Full-length Loop Genomics sequencing technology; PacBio:
Full-length the PacBio single molecule, real-time (SMRT) technology.

Fig. 2. Taxonomy profiles in different sequencing methods datasets. The proportion of annotation
sequences from the V4 ($n=48$, blue), LoopSeq ($n=48$, yellow), and PacBio ($n=48$, orange) datasets
was determined by comparing the sequence with the SILVA database, and are represented at the (A)
phylum, (C) genus, and (E) species levels. Venn diagram showing the numbers of unique and shared
(B) phylum, (D) genus, and (F) species between three sequencing methods. Blue denotes V4, yellow
denotes LoopSeq, and orange denotes PacBio.

Fig. 3. Relative abundance analysis of common genus in three sequencing methods. The most
abundant 30 genera were selected from shared 201 genera of the three sequencing methods. Color
pairs denote samples of three sequencing methods in neutral or acidic soil with different waterlogging
851 times. Bubble sizes indicate the read abundance of an individual genus.

Fig. 4. Rhizosphere core microorganisms of different sequencing methods. The different parts inside
the double pie chart represent the bacterial phyla of the soybean core microbiome. The different parts
outside the double pie chart represent the OTU (genus) of the soybean core microbiome, and each
OTU (genus) is assigned to the corresponding bacterial phyla. The size of the different double pie

chart portions represents the percentage of phylum/genus relative abundance in all core microbial
components.

Fig. 5. (A, B, C) Network analysis reveals the symbiotic pattern between OTUs. The nodes are colored
according to the modular type. The connections between nodes indicate strong and significant
(Spearman's $r > 0.8$ or $r < -0.8$) ($P < 0.01$) correlation. The volcano map shows the amount of OTU
enriched and depleted in neutral soil and after waterlogging in the modules of different sequencing
methods, respectively. Violet denote Module I, green denotes Module II, blue denotes Module III,
black denotes Module IV, orange denotes Module V, red denotes Module VI, cyan denotes Module VII,
grey denotes other modules.

Fig. 6. Paired comparison of environmental factors and microbial community with a color gradient
denoting Pearson's correlation coefficient. Spearman's correlation coefficient > 0 indicates positive
correlation and < 0 indicates negative correlation. Effects of environmental factors in (A) two types
of soil, (B) neutral soil, (C) and acidic soil on the microbial communities of the three sequencing
methods. The edge width corresponds to the distance dependence of Mantel's R statistic, and
Statistical significance based on 9,999 permutations represents edge color. Mantel's r size indicates the
strength of the correlation. The color of the connecting line indicates the correlation between different
sequencing methods and environmental factors.

B

LoopSeq

C

PacBio

A

V4

- Module I (35.88%)
- Module II (32.76%)
- Module III (28.45%)
- Module IV (0.83%)
- Module V (0.35%)
- Module VI (0.21%)
- Module VII (0.21%)
- Other modules(1.31%)

B

LoopSeq

- Module I (32.98%)
- Module II (14.61%)
- Module III (5.42%)
- Module IV (4.67%)
- Module V (1.96%)
- Module VI (1.81%)
- Module VII (1.81%)
- Other modules(36.82%)

C

PacBio

- Module I (28.82%)
- Module II (22.94%)
- Module III (20.59%)
- Module IV (20%)
- Module V (1.18%)
- Module VI (1.18%)
- Module VII (1.18%)
- Other modules(4.11%)

Dear editor and reviewers,

Thank you for considering our manuscript (Spectrum02011-21) entitled “Effects of waterlogging on soybean rhizosphere bacterial community using V4, LoopSeq and PacBio 16S rRNA sequence”. We are grateful for the constructive feedback provided by you and two reviewers which have greatly improved the quality of the manuscript.

We have studied the comments carefully and have made corrections which we hope meet with your approval.

We would be grateful if our revised manuscript, with your kindest approval, can be accepted for publication in the *Microbiology Spectrum*. Please contact me if you have any further questions.

Best regards,

Tengxiang Lian

Email: liantx@scau.edu.cn

Reviewer #1 (Comments for the Author):

Yu et al. investigated the effects of waterlogging and soil type on soybean rhizosphere bacterial structure using three strategies of partial (V4) 16S rRNA sequencing and full-length sequencing of LoopSeq and PacBio. Their results showed that waterlogging significantly influenced the bacterial community structure, with increased relative abundance of *Geobacter*. However, they also stated that whether waterlogging increased the relative abundance of the nitrogen or phosphorus relevant microbes, were dependent on the sequencing methods used according to the co-occurrence network analysis. Be surely, treatment would result in different microbial enrichment which are critical for the stability of bacterial community structure and ecological functions. Altogether, the authors believed that their findings highlight the microbial roles in assisting soybean to resist stress under waterlogging condition, and they confirmed that LoopSeq method could improve the accuracy at species-level environmental samples, and was also cheaper than other two sequencing techniques. Overall, I feel that this study is well organized and has proposed bright viewpoints; the data analyses seem to be solid and the manuscript is well written. I have two suggestions: 1) please give more details about the sequence processing which would make the reader clearer about what happen (e.g. reads number changes) during each step, these new information would be added in the MM or Results sections, and 2) I was thinking

that 16S sequencing would get some of archaeal sequences which were removed or not in this study, if not, mentioning 'bacterial' would be improper, alternatively, using 'prokaryotic' may be more appropriate.

Response: Thank you very much for your suggestion.

(1) We have added more details of the about the sequence processing to the material based on the suggestions. Such as line 197-204.

(2) It is true that 16S sequencing would get some of archaeal sequences. In this study, the main objective was to explore the effects of waterlogging stress on bacterial structure using the three sequencing methods. Therefore, we removed the information of archaea from the sequence for the subsequent analysis. We added the description to the MS as follow:

Line 199-200 “The information of archaea was removed from the sequence for the subsequent analysis.”

Reviewer #2 (Comments for the Author):

SUMMARY

In this manuscript, the authors test the effects of waterlogging on rhizosphere bacterial communities of two different types of soybean plants (resistant or susceptible to waterlogging), in two different soils (acidic or neutral pH), using three different DNA amplification/sequencing methods (16S V4 PCR/Illumina, 16S full-length PacBio, and 16S full-length LoopSeq).

They had three hypotheses: 1a) waterlogging will decrease bacterial diversity in the soybean rhizosphere and 1b) the effect would be stronger in the acid soil than the neutral soil; 2) taxonomic resolution of microbial communities would be similar in both full-length sequencing technologies, and lower in V4 sequencing; 3) specific bacteria would be enriched in the rhizosphere that may help soybeans resist waterlogging stress.

Addressing these hypotheses, the authors show that:

1. waterlogging had no significant ($p < 0.05$ one-way ANOVA) effect on alpha-diversity except for a lower value (Chao1) in acidic soil with Loopseq.
2. Taxonomic resolution of sequenced communities were distinct in each of three amplification/sequencing technologies when compared to the same database, with slightly more (~97% vs ~95.5%) V4/Illumina sequences assigned to phylum than either LoopSeq or PacBio, but far less

assigned to genus (~45% vs. ~88-96%) or species (~15% vs. 30-95%). All assigned proportions are distinct from each other when every sample is regarded as independent (n=48).

3. Some bacterial OTUs found in both neutral and acidic soils (60 with V4, 10 with LoopSeq, 9 with PacBio) were relatively more abundant (differential expression analysis) under waterlogging than in control soils.

The authors further show that microbial beta-diversity (CPCoA) was different in each soil and each treatment, but with distinct results dependent on amplification/sequencing technology. Furthermore, that the identities of organisms were drastically different between amplification/sequencing technologies, with the starkest examples being in phylum Acidobacteria (~1/3 as abundant from PacBio compared to either V4/Illumina or LoopSeq) and Planctomycetes (4-5x more abundant in PacBio compared to either V4/Illumina or LoopSeq). The authors discuss the fact that choice of amplification/sequencing method alters the results and interpretation of their waterlogging test; some potentially helpful microbes only appear with differential abundance using some methods. Others (*Geobacter*) appear with all methods. The authors use changes of relative abundance of members of specific modules determined with network analysis to further explore whether waterlogging increased specific microbes that may contribute to helping soybeans resist flooding stress. Without identifying the mechanisms by which these specific microbes do protect soy plants, or even that they do, the authors recognize that though it is reasonable to hypothesize that these microbes may help protect soy plants, this study does not provide evidence that they actually do. The authors conclude with a statement that LoopSeq is a cheaper and higher-utility option than PacBio for full-length 16S sequencing.

REVIEW

I commend the authors for their excellent work! I believe the data and analyses presented in this manuscript would be of immediate and substantial use to many microbial ecologists. However, there are critical issues that must be addressed, mostly omissions of methods and some expected results. These are presented in detail below. In addition, I have a broad suggestion for a re-framing of the data that I would ask the authors consider, but this is only my opinion, and not required for publication.

I have split my comments into three sections: 1) major issues that require amendment, 2) minor edits,

and 3) a broad (and totally optional) suggestion for substantial change to the framing of the data and conclusions. The final suggested change would only be to frame the data already presented in this manuscript in a way that I believe may help a wider audience of microbial ecologists engage with it; it is not required in any way.

Response: Thank you very much for your recognition of our work, and we also appreciate you thank you making such detailed suggestions, which greatly increase the reproducibility and rigour of the MS. We have carefully revised and answered each amendment and suggestion and hope you will approve.

Major Amendments

Line 109 hypotheses ii and iii are well-developed by the previous paragraphs in the introduction, but so far no information has been presented to suggest why waterlogging would decrease bacterial diversity in the soybean rhizosphere, nor why this should be more severe in acidic soil. Either support the first hypothesis in the introduction, or omit this hypothesis.

Response: Thank you for the suggestion. To make our first hypothesis more reasonable, we added a description of the effect of waterlogging on microbial diversity. Moreover, we removed the hypothesis that microbial diversity is comparable between acidic and neutral soils.

Line63-65. “In particular, soil physicochemical properties (e.g., porosity, structure, and pH) will be the worst affected, where a sharp decrease in soil oxygen concentrations will have a negative impact on microbial diversity and community activity (7, 8).”

Line110-111 “We hypothesized that (i) waterlogging will decrease the bacterial diversity in the soybean rhizosphere,”

Line 117 more information is required regarding the soil sampling and preparation. First, soil classification, as detailed as possible, according to any system of soil taxonomy (preferably World Reference Base) is needed for each soil. Second, the approximate sampling depth in cm, plus the number and size of soil samples taken, is needed. Together these are needed to consider the native habitats of the soil microbes in this study. If the soil were taken from the top 5-10 cm with a strong structure, it is possible that these microbes are never waterlogged for long, and are relatively unadapted to flooding. If the soil were taken from deeper horizons (30+ cm) it is possible that the microbes are relatively well-adapted to hypoxia and waterlogging. In the latter case, it may be the non-waterlogged soils that represent the more intense disturbance from the native habitat. Additionally, it is later stated

that the soils were sieved and air-dried but this procedure should be stated here. The process of sampling, sieving, and drying soil up to when it was put in pots should be detailed to the extent that it could be replicated.

Response: Thank you very much, we added more information of the soil sampling and preparation.

Line117-121. “A neutral and an acid soil were collected at the surface layer with a depth of 10 cm from Yingde County (113°40’N, 24°18’E), and Suixi County (110°25’N, 21°32’E), Guangdong Province, China, respectively. 500 kg of each type soil were then air-dried for 5 days and sieved through a 2-mm mesh to remove impurities before planting the soybeans. The neutral and an acid soil were classified as Kanhaplohumults and Paleustults, respectively, according to USDA soil taxonomy.”

Line 118 more information is required about the fate of these two soybean varieties. Despite the fact that two soybean varieties were used, they are never mentioned again throughout the manuscript, though their microbiomes were analyzed. Whether or not these two varieties' microbial communities differed, they should be mentioned again in the results. If they did differ significantly and this is relevant to the conclusions presented in this study, they should be presented and discussed. If they differed significantly but this did not affect the conclusions presented in this study, that should be stated in the results, with the reasoning for not considering soybean variety as an experimental factor explained in the text. If they did not differ significantly, that too should be stated in the results, with that rationale given for why they were treated only as a blocking factor in the statistics (if they were - many tests are given as n=6 so I assume they were not treated as independent soy replicates).

Response: Thank you for the suggestion.

Apologies for the mistakes we have made here. We designed two experiments at the same time. One experiment was grown with two soybean varieties, in order to explore the structure and function of microorganisms recruited by different soybean varieties. However, the main objective of this study was to explore the effects of waterlogging on soybean rhizosphere microorganisms on two types of soil, and to compare the differences among the different sequencing methods. Therefore, we did not consider varieties here, but mixed the two varieties as replicates for the subsequent analysis. Each treatment should be 12 replicates. Again, we are very sorry for our carelessness. In addition, we thank the two professors who provided us with the soybean varieties in the acknowledgements.

Line 121-124 “Two soybean varieties, i.e., Qihuang34 and Jidou17, that are widely grown in central

and southern China, were used in this study. Soil physicochemical properties and rhizosphere microorganisms of both varieties were mixed as replicates for the subsequent analysis.”

Line547-548. We are very grateful to Prof. Xu Ran and Prof. Zhang Mengchen for providing the soybean varieties.

Line 122 more detail and structure is needed in this paragraph to get an accurate sense of the experimental design. From what I understand from this paragraph, 6 pots were prepared for each of two soil types, soybean types, and water treatments ($6 \times 2 \times 2 = 48$ pots). 24 pots were filled with neutral soil, 24 pots were filled with acid soil. Eight seeds of either one or the other soybean type were added to each pot. After 6 days past emergence, the number of plants in each pot was thinned down to three. At the V2 growth phase, water was added to the pots until standing 4-6 cm above the soil for waterlogging treatment, while no additional water was added to the control treatment. If this is correct, then all the information is here but the paragraph should be written in a way that would allow for replication: 1) state the number of replicates, experimental factors, and total pots together in the second sentence 2) state how the three remaining soybean plants were selected to remain, while others were removed (Healthiest? Furthest spaced? Random?) 3) State why waterlogging stress examination was performed and what was done with that information 4) state when the water was added (at the V2 stage, right after stress examination?) 5) state how long waterlogging treatment persisted; that is, how long the soils remain saturated. In the following paragraph, it is stated that rhizosphere soil was collected after three days of waterlogging treatment - does that represent three days with 4-6 cm standing water in the pots for all three days? If the soils did drain quickly or it is impossible to know how long the soils were saturated, mention this here.

Response: Thanks. We have changed this paragraph to a more detailed description based on your suggestion.

Line127-140. “We conducted a random pot experiment in the greenhouse at the Agricultural College of South China Agricultural University, Guangzhou, China. The experiment was a completely randomized block design. In total, 48 pots of soybean suffering or not from waterlogging in two types of acidic soil (2 soil types \times 2 waterlogging treatments = 4 treatments) were conducted. For each treatment, there were twelve pots replicates (6 replicates for each variety) with 4 seedlings per replicate. Each pot (top diameter 13.8 cm, bottom diameter 10.4 cm, height 12.2 cm) used in this study contained about 2.5 kg

of air-dried soil. Eight strong and full soybean seeds with similar shapes were sown in each pot. The soybean growth process was carried out in a greenhouse with controllable conditions (temperatures of 26-32 °C in the daytime and 15-21 °C in the nighttime). After 6 days of emergence, 3 healthy soybean plants with the same growth were left. Waterlogging stress examination was performed at the soybean in the V2 stage, in order to explore three sequencing methods for waterlogging-affected soybean rhizosphere microbes. For waterlogging treatment, the water was added to the pots up to 4 to 6 cm above the soil, and more water was added to the soil twice a day for three days to ensure the water level. For the control plants, the water content were left in an ideal environment.”

Line 135 more detail is required regarding soil sampling. How was the rhizosphere soil collected? There are different definitions of what qualifies as 'rhizosphere'. It is implied here that the entire root network of all three soybean plants in each pot were combined in one 50 mL tube for each pot, the roots in the tube was centrifuged, and 5g of soil collected from the bottom of the tube was collected for DNA extraction. If that is true, state the order in which these things were done with sufficient detail (for example how rhizosphere soil was designated, centrifugation speed) such that another lab could replicate it.

Response: Thanks. We have shown the sampling process in detail in this paragraph.

Line143-150. “After three days of waterlogging treatment, the bulk soil was removed from the root by manually shaking, and then the entire root of all three soybean plants in each pot were transferred into a 50 ml centrifuge tube filled with phosphate-buffered saline (PBS) to collect the rhizosphere soil (defined as the soil that adheres to the root). After that, the centrifuge tubes were placed on a shaker (120 rpm/min, 25 °C) for 20 min, and then centrifuged for 10 min (6,000 × g, 4 °C). 5 g of deposited rhizosphere soil was collected from each sample and placed in a sterilization centrifuge tube for storage at -80 °C for DNA extraction. The remaining rhizosphere soil was stored at 4 °C prior to the determination of soil physical and chemical properties.”

Line 142 more detail is required regarding soil analyses. How long was rhizosphere soil stored at 4°C prior to analysis? Was rhizosphere soil dried and sieved prior to analysis? Either describe pH measurements and preparation and extraction of soil solution for TN, TK, SOC, NH₄, NO₃, TP, and Olsen-P in sufficient detail to replicate these analyses, or cite extraction methods here. Additionally,

these data are never reported; please include a table of mean soil data values +/- standard errors for each treatment group in the supplementary material.

Response: Thanks. We have shown the soil physical and chemical properties analyses process in detail in this paragraph and shown the soil analyses results in the supplementary material (Table S1).

Line152-161. "After sampling one week, the soil was air-dried for 5 days and sieved through a 2-mm mesh to remove plant residues. Then, soil physical and chemical properties were measured according to previous studies (33-34). In general, soil pH was determined using a pH meter (FE20-FiveEasy™ pH, Mettler Toledo, Germany) in soil water suspension (5:1 water-to-soil). Total nitrogen (TN) was determined using an Ultraviolet Spectrophotometer (UV-1800, Suzhou, China). Total potassium (TK) in soil was measured by flame atomic absorption spectrometer (AA-7000, Shimadzu, Japan). Soil organic carbon (SOC) content was assessed using a TOC-5000A analyzer (Shimadzu, Kyoto, Japan). The content of NH_4^+ , NO_3^- , TP, and Olsen-P in soil was determined by a continuous flow analytical system (SKALAR SAN++, Netherlands). The effect of waterlogging on soil chemical properties was summarized in Table S1."

Line 177 Unless DADA2 is implemented very differently in QIIME2 than other platforms, DADA2 algorithm produces ASVs to stand in for bacterial organisms, which are distinct from OTUs. This is significant as the difference between ASVs and classified OTUs could account for some of the difference in taxonomy between these amplification/sequencing technologies, especially between LoopSeq (ASVs) and PacBio (OTUs) despite the fact that both are long-read technologies. Clarify in the text when organisms are represented by ASVs vs. OTUs.

Response: Thanks. It's true that DADA2 algorithm produces ASVs. In this study, in order to be consistent with OTUs produced by UPARSE algorithm, and to be more concise in the description below, we replaced the ASVs with an OTUs. Moreover, we added the corresponding descriptions as follow:

Line197-199. "In order for ASVs produced by DADA2 algorithm consistent with OTUs produced by UPARSE algorithm, and to be more concise in the description below, we replaced the ASVs with OTUs.

Line 198, 298, 432 The authors perform network analysis on microbial communities to analyze

co-occurrence between microbes. The authors cite Faust and Raes 2012 to introduce and interpret their network analysis results, but Faust later published a cautionary manuscript demonstrating that the assumptions under which network analysis works are often violated, and the possible inferences overstated, in soil microbial ecology ('From hairballs to hypotheses-biological insights from microbial networks', Rottjers and Faust, 2018). Furthermore, choice of network analysis method and network size can affect network accuracy and meaningfulness, even when assumptions are met ('Difficulty in inferring microbial community structure based on co-occurrence network approaches', Hirano and Takemoto, 2019). The authors must consider at least Rottjers and Faust, 2018 and address the limitations of network analysis within the context of this study, in the discussion section. This is needed to allow readers to fully evaluate the results presented.

Response: Thanks. We have added a more discussion based on your suggestion.

Line488-493 “It is worth noting that the conclusions we obtained using network analysis were speculative and cannot be taken as definitive information. This is because that networks provide a valuable tool, but they are best seen as hypothesis generators rather than as a solid conclusion. This is because these methods only infer ecological associations. Furthermore, the choice of network analysis method and network size can affect the accuracy and significance of the network, even when assumptions are met (96, 97)”

Line 216 More information is required about the sequencing results. What sequencing depth was obtained for your samples in each of three sequencing methods (minimum, maximum, and mean)? Were sequenced communities rarefied? If so, how; if not, how was difference in sequencing depth corrected for in diversity analyses?

Response: Thanks. We rarefied the sequencing results of the three sequencing methods.

Line200-204. “In total, 1,716,803, 168,704 and 214,645 bacterial 16S rRNA high-quality reads were obtained from 48 samples, with an average of 35,766, 3,551 and 4759 reads per sample after rarefied for the three sequencing methods, respectively. These read were sorted into 7,673, 6,494and 38,106 OTUs of the three sequencing methods, respectively, for the subsequent analysis.”

Line 218 There must be more coherence between statistics as presented in the methods section, results section, the figure captions, and results shown in the figures. For instance, it is stated in the

methods section (line 189) that a three-way ANOVA was performed to assess the difference in bacterial alpha-diversity. It is stated in the results (line 218) that waterlogging had no significant impact on bacterial alpha-diversity (except for acidic soil sequenced by loopseq) - one-way ANOVA, $n=6$, $p > 0.05$. In the caption for this figure (line 830) it states "Effects of waterlogging and soil types on soybean rhizosphere... one-way ANOVA, $n=6$, $p < 0.05$). In the figure itself, letters denoting significant differences correspond to none of these three apparently distinct tests, instead showing the results of all pairwise comparisons, which are never explained in the text. The authors must describe all post-hoc pairwise comparison tests they use in the methods section, with multiple-comparison correction factors, and should discuss significant interaction terms in a previously-conducted 3-way ANOVA rather than calculate a new post-hoc 1-factor ANOVA for each individual comparison of interest.

Response: Thank you for the suggestion, we have added the 2/3-way ANOVA here, It also showed all the analysis methods used in the article. Moreover, added the results and discuss significant interaction terms in a 2/3-way ANOVA and then compared the differences between the different treatments (Table S1 and Table S2).

Line208-213 “A three-way analysis of variance (ANOVA) and Dunn’s multiple comparison with Bonferoni correction were used to identify significant differences in classification resolution, bacterial alpha diversity and the relative abundances of bacterial phyla and genera among all treatments. Moreover, a two-way ANOVA and multiple comparison were used to identify the significant differences in soil chemical properties”

Line 237-239 “The sequencing methods, waterlogging, soil type and their interaction had a significant effect on alpha diversity (Table S2). Higher alpha diversity was obtained with V4 sequencing than with full-length sequencing. In more detail, the...”

Line 253-254 “The results showed that the sequencing methods, soil types and their interactive significantly affected the annotation proportion at low classification levels (such as species level) (Table S2).”

Line 360-362 “The two-way ANOVA revealed that waterlogging, soil types, and their interactive significantly affected some soil physicochemical properties. For example, waterlogging significantly affected the Olsen-P, NH_4^+ and NO_3^- , and their interactive significantly affected the NO_3^- (Table S1).”

Minor Edits

Line 32 replace "cycle" with "cycling"

Response: Line32, 53, 450, 467, 481, 485, 487, 498, 537. Thanks. Revised.

Line 34 replace "enriched" with "enriched members"

Response: Line34. Thanks. Revised.

Line 87 replace "biasness" with "bias", throughout the manuscript

Response: Line88, 98, 233, 414. Thanks. Revised.

Line 230 the captions on Fig. S1 and S3 are switched.

Response: Thanks. Revised.

Line 244+ only genus and species are italicized, all other taxonomic ranks are not.

Response: Line270-278, 308-322. Thanks. Revised.

Line 311 please reference figure 5 again when discussing the volcano plots here. Also - these volcano plots are interesting and important within this manuscript but are almost too small to be interpreted - especially if one were reading the manuscript in print, one would have to take the authors' word for the number and position of ASVs and OTUs involved, since they are too small to be read. I'm not sure if these could be expanded in a separate figure, but the current arrangement is not ideal.

Response: Thanks. we have resized the volcano plots.

Line 498 please review the references, multiple papers are cited separately multiple times and there are some typos in the paper titles.

Response: Thanks. We reviewed the references in order to avoid duplication of citations.

Line 826 please define Ne, Ac, W, and CK in this caption. Additionally, consider defining 'CK' in the text - W for Water is self-explanatory but I'm not sure what CK stands for.

Response: Line900. Thanks. Revised.

Line 851 state how the read abundances were calculated for this figure - averaged across every sample?

Response: Line917-919. Thanks. "Bubble sizes indicate the averaged relative abundance of an individual genus across each treatment."

Finally, there are many very minor errors in grammar and syntax which don't interfere with the meaning of the text at all, but the authors may wish to get a copy-editor to check it over as a matter of style.

Response: Thanks. We have carefully revised the grammatical errors in the manuscript based on your suggestion

Framing - SUGGESTION ONLY, NOT REQUIRED FOR PUBLICATION

The manuscript as written puts first emphasis on the ecological experiment conducted: the effects of water saturation on soy rhizosphere communities in two distinct soils. Throughout the manuscript, the importance of understanding the effects of waterlogging on rhizosphere microbial communities is evaluated before the consideration of amplification and sequencing technology. There is a deep literature on the effects of saturation on soil microbial communities and traits, as the authors cite in their discussion (Evans and Wallenstein 2012 etc). It seems that the authors include the comparison of the three methods (V4-Illumina, most commonly used by microbial ecologists; PacBio, commonly recognized as superior for its longer read length but uncommonly used because of its cost and historically high error rate; and LoopSeq, uncommonly used because of its novelty) as a secondary component of the manuscript. Though the authors' experimental results are certainly interesting and useful to those studying soil microbial adaptations to moisture stress, the exploration of the ways in which selection of a DNA amplification/sequencing technology alters their results is applicable to all microbial ecologists evaluating their own or others' V4 Illumina data, or considering long-read sequencing for future projects.

As the authors say, the field of microbial ecology is significantly based on Illumina V4 sequencing. Though there are many issues with the technology, it is widespread because there is a consensus among microbial ecologists that it is 'good enough' for community profiling. The question I asked myself as I read this manuscript is: would selection of the 'wrong' sequencing technology lead to incorrect results even from a well-constructed study? The authors provide an excellent test of this question, showing exactly how different methods produce slightly different answers to an important question in microbial ecology, though they do not explicitly frame it this way. Instead, the authors tend to focus first on the results of their experiment across all sequencing methods, then only secondly discuss ways in which the method changed the results, if they are mentioned in the text. Even when the differences between sequencing methods are directly addressed, they are often presented as being three equally-valid ways of looking at the community, rather than three different methods, some of which work better than others, that all purport to observe one 'true' community. I believe different results derived from these

methods should be cause for interest, analysis, and alarm among microbial ecologists - they cannot all be correct.

The following are four specific examples of what I perceive as the authors' emphasis on the ecological side of these results over the methodological side, just to give a better sense of what I mean:

1. In the introduction, the authors discuss the issue of crop damage from waterlogging and the possibility that specific microbes may be able to reduce crop damage from waterlogging first (paragraphs 1 and 2) before discussing the issues of lower resolution and inaccurate identification of microbial communities in short-read compared to long read sequencing (paragraphs 3 and 4). However, the authors can and do definitively answer the questions raised by comparing sequencing methods, while they can't address the question of whether any of the microbes they saw actually protect soybeans against waterlogging.

Response: Thanks. We have added a more detailed description based on your suggestion.

Line468-472. “Nevertheless, whether the core microorganisms we discovered could establish a defense mechanism against waterlogging damage with soybeans is still unclear. To address these issues, the use of high-throughput cultivation and identification of microbes (86) as well as synthetic communities (87) can help explore the extent to these recruited microorganisms contribute to soybean resistance to waterlogging stress.”

2. In figure 1A, the authors show that both long-read sequencing methods produced communities with Chao1 alpha-diversity ~250, which did not differ by soil type, compared with the short-read sequencing method Chao1 alpha-diversity results of 500-600, which showed significantly more alpha-diversity in neutral soils than acidic soils. The authors have already established that shorter reads lead to less accurate classification of OTUs (lines 88-91); a reasonable conclusion might be that had a researcher relied on V4, they would have come to the (likely false) conclusion that soil type significantly influenced bacterial alpha-diversity, whereas with better sequencing methods (either PacBio or LoopSeq), they would have accurately found that soil type made no (or context-dependent) difference. Instead, the authors only report that there was no overall negative effect of waterlogging on soybean microbe diversity, contrary to their first hypothesis.

Response: Thank you for reminding us that we do neglect to discuss the results of different sequencing methods. Therefore, we have added the following discussion.

Line378-389. “In this study, we used three sequencing methods to evaluate the impact of waterlogging on the structure of soybean rhizosphere microbial communities on two types of soil. Our first hypothesis was not verified, as the results from three sequencing methods showed that waterlogging had no significant impact on the bacterial alpha diversity (Fig. 1AB). However, V4 sequencing, but not full-length sequencing, showed significantly more alpha-diversity in neutral soil than in acidic soil. A reasonable conclusion might be that if the researchers had relied on V4 sequencing, they would have concluded (probably incorrectly) that soil type clearly influenced bacterial alpha-diversity, whereas with a better sequencing method (PacBio or LoopSeq), they would have accurately found no (or context-dependent) differences in soil types. For the other two hypotheses, they were fully verified, that the resolution of PacBio and LoopSeq were significantly higher than partial sequencing, and some beneficial microorganisms, such as *Geobacter*, were enriched in the soybeans rhizosphere that may help soybeans resist to waterlogging stress.”

3. The effect of amplification/sequencing method on microbial beta-diversity, though seemingly extreme, is not directly compared in Figure 1, which focuses on the effects of soil type and waterlogging instead (Fig. 1C-E, figure S3). How distinct are these communities between sequencing methods, if you plot them on the same CPCoA? And interestingly, when it comes to beta-diversity, the long-read methods do not produce very similar answers (as they did for alpha-diversity) - for some phyla (Proteobacteria) abundances are broadly similar across all methods, for some (Planctomycetes) PacBio yields very different results than either V4 Illumina or LoopSeq, and for some (Acidobacteria) all three methods are distinct. The effect of waterlogging on beta-diversity is mentioned promptly in the discussion (line 356), but the effect of sequencing method on beta-diversity is not mentioned, and the reader has to find it in the supplementary materials.

Response: Many thanks. This is really an excellent suggestion. We have previously considered three sequencing methods for total PCoA analysis. However, total PCoA analysis is difficult when using different OTUs from all three sequencing methods. This is because the three methods produce separate OTU table corresponding to different taxonomic annotations. For example, a certain genus corresponds to V4, Loop and PacBio may be OTU1, OTU13 and OTU225, respectively. If we put such OTUs together for analysis, we will get the wrong information. Therefore, we were not able to do a total

PCoA. However, we appreciate that you have made such a great suggestion, and we will consider it carefully in the future work and try to find a solution to this problem.

4. In the discussion, the authors bring up the question of whether they would have observed different results had they used one sequencing method or another (line 381) - but imply by omission throughout the discussion that all three methods are equally valid; that is, that the significant difference in Variovax detected in V4 sequencing is just as accurate and useful a finding as the increase in *Pirellula* detected with PacBio. Earlier in this manuscript the authors show that genus-level resolution is far higher in LoopSeq and PacBio than in V4 Illumina, and that species-level resolution is much higher than both in PacBio; why then are results from these three sequencing methods all presented as approximately equally valid and useful?

Response: Thanks. We have added a more detailed description based on your suggestion.

Line423-427. “These results indicated that there were differences in the information about the recruited bacteria detected by different sequencing methods. Due to the high resolution of full-length sequencing, we may conclude that the use of V4 sequencing may not give us very accurate information about the recruited microorganisms, which might affect our screening of waterlogging tolerance-related microorganisms.”

There are other manuscripts in microbial ecology that use both short-read and long-read sequencing with LoopSeq (<https://doi.org/10.3389/fmicb.2021.598180>) but none that I can find has demonstrated in a way that this manuscript does, how the conclusions of a study might change depending on these sequencing methods used - not just the accuracy of the methods by themselves in reconstructing a mock community, but the way in which the actual results of an experiment do or don't change between methods (for instance, *Geobacter* was detected across methods, other microbes were not). In my opinion, the comparison between amplification/sequencing methods might be due further explicit discussion within the manuscript. Having said this, all these conclusions can be made from a close reading of the data already presented in the manuscript, and the authors do show and explain these differences within the figures and discussion already. The authors should feel free to ignore the comments and suggestions in this section if they disagree with my interpretation.

Response: Thanks. We have added more detailed discussion based on your suggestion.

Line520-529. “The non-negligible difference of the bacterial diversity, comparisons at the level of

bacterial phylum and genera, core microorganisms, network co-occurrence analysis and correlation with environmental factors was in line with that the sequencing bias should be taken into account by different sequencing method for microbial community analysis (106, 107). Our results are consistent with previous findings that while short-read sequencing is effective in microbiome analyses at higher taxonomic levels (e.g. phylum level), LoopSeq and PacBio analyses show greater power to delve deeper into taxonomic capabilities (24, 30). We cannot deny that the conclusions or predictions of previous studies based on V4 sequencing may be indicative of importance, but at the least in this study, in which both full-length sequencing methods showed similar results and identified more functional microorganisms than that in V4 sequencing.”

January 16, 2022

Dr. Tengxiang Lian
South China Agricultural University
Guangzhou
China

Re: Spectrum02011-21R1 (**Effects of waterlogging on soybean rhizosphere bacterial community using V4, LoopSeq and PacBio 16S rRNA sequence**)

Dear Dr. Tengxiang Lian:

Your manuscript has been accepted, and I am forwarding it to the ASM Journals Department for publication. You will be notified when your proofs are ready to be viewed.

Sincerely,

Cheng Gao
Editor, Microbiology Spectrum

**Effects of waterlogging on soybean rhizosphere bacterial community using V4, LoopSeq and**
**PacBio 16S rRNA sequence**

Taobing Yu^{a,b}, Lang Cheng^{a,b}, Qi Liu^{a,b}, Shasha Wang^{a,b}, Yuan Zhou^c, Hongbin Zhong^c, Meifang Tang^c,
Hai Nian^{a,b,*}, Tengxiang Lian^{a,b,*}

5 ^a*The State Key Laboratory for Conservation and Utilization of Subtropical Agro-bioresources, South*
*China Agricultural University, Guangzhou 510642, Guangdong, People's Republic of China*

7 ^b*The Key Laboratory of Plant Molecular Breeding of Guangdong Province, College of Agriculture,*
*South China Agricultural University, Guangzhou 510642, Guangdong, People's Republic of China*

9 ^c*BGI Genomics, BGI-Shenzhen, Shenzhen 518083, China*

* **Corresponding author1:** Tengxiang Lian

**Corresponding address:** No.483 Wushan Road, Guangzhou, Guangdong, 510642, China.

**Tel:** +86 02085288024

Fax: +86 02085288024

E-mail address: liantx@scau.edu.cn

* **Corresponding author2:** Hai Nian

**Corresponding address:** No.483 Wushan Road, Guangzhou, Guangdong, 510642, China.

**Tel:** +86 02085288024

Fax: +86 02085288024

**E-mail address:** hnian@scau.edu.cn

[revised manuscript text omitted]

soybeans rhizosphere that may help soybeans resist to waterlogging stress.

**2. Methods and material**

**2.1. Soil and soybean material**

A neutral and an acid soil were collected at the surface layer with a depth of 10 cm from Yingde
County (113°40'N, 24°18'E), and Suixi County (110°25'N, 21°32'E), Guangdong Province, China,
respectively. 500 kg of each type soil were then air-dried for 5 days and sieved through a 2-mm mesh to
remove impurities before planting the soybeans. The neutral and an acid soil were classified as
Kanhaplohumults and Paleustults, respectively, according to USDA soil taxonomy. Two soybean
varieties, i.e., Qihuang34 and Jidou17, that are widely grown in central and southern China, were used
in this study. Soil physicochemical properties and rhizosphere microorganisms of both varieties were
mixed as replicates for the subsequent analysis (31, 32).

**2.2. Experimental design**

We conducted a random pot experiment in the greenhouse at the Agricultural College of South
China Agricultural University, Guangzhou, China. The experiment was a completely randomized block
design. In total, 48 pots of soybean suffering or not from waterlogging in two types of acidic soil (2 soil
types × 2 waterlogging treatments = 4 treatments) were conducted. For each treatment, there were
twelve pots replicates (6 replicates for each variety) with 4 seedlings per replicate. Each pot (top
diameter 13.8 cm, bottom diameter 10.4 cm, height 12.2 cm) used in this study contained about 2.5 kg
of air-dried soil. Eight strong and full soybean seeds with similar shapes were sown in each pot. The
soybean growth process was carried out in a greenhouse with controllable conditions (temperatures of
26-32 °C in the daytime and 15-21 °C in the nighttime). After 6 days of emergence, 3 healthy soybean
plants with the same growth were left. Waterlogging stress examination was performed at the soybean
in the V2 stage, in order to explore three sequencing methods for waterlogging-affected soybean
rhizosphere microbes. For waterlogging treatment, the water was added to the pots up to 4 to 6 cm
above the soil, and more water was added to the soil twice a day for three days to ensure the water level.
For the control plants, the water content were left in an ideal environment.

**2.3. Soil sampling**

After three days of waterlogging treatment, the bulk soil was removed from the root by manually
shaking, and then the entire root of all three soybean plants in each pot were transferred into a 50 ml
centrifuge tube filled with phosphate-buffered saline (PBS) to collect the rhizosphere soil (defined as
the soil that adheres to the root). After that, the centrifuge tubes were placed on a shaker (120 rpm/min,
25 °C) for 20 min, and then centrifuged for 10 min (6,000 × g, 4 °C). 5 g of deposited rhizosphere soil
was collected from each sample and placed in a sterilization centrifuge tube for storage at -80 °C for
DNA extraction. The remaining rhizosphere soil was stored at 4 °C prior to the determination of soil
physical and chemical properties.

**2.4. Analysis of soil properties**

After sampling one week, the soil was air-dried for 5 days and sieved through a 2-mm mesh to
remove plant residues. Then, soil physical and chemical properties were measured according to
previous studies (33-34). In general, soil pH was determined using a pH meter (FE20-FiveEasy™ pH,
Mettler Toledo, Germany) in soil water suspension (5:1 water-to-soil). Total nitrogen (TN) was
determined using an Ultraviolet Spectrophotometer (UV-1800, Suzhou, China). Total potassium (TK)
in soil was measured by flame atomic absorption spectrometer (AA-7000, Shimadzu, Japan). Soil
organic carbon (SOC) content was assessed using a TOC-5000A analyzer (Shimadzu, Kyoto, Japan).
The content of NH_4^+ , NO_3^- , TP, and Olsen-P in soil was determined by a continuous flow analytical
system (SKALAR SAN++, Netherlands). The effect of waterlogging on soil chemical properties was
summarized in Table S1.

**2.5. DNA extraction from soil samples and sequencing process**

Total soil DNA was extracted using a Fast DNA SPIN Kit for Soil (MP Biomedicals, Santa Ana,
CA) following the manufacturer's recommendations. The DNA was eluted with 80 µl H_2O , and
analyzed by Nanodrop 2000 spectrophotometry. Primers 515F (5'-GTGCCAGCMGCCGCGGTAA-3')
and 806R (5'-GGACTACHVGGGTCTAAT-3') with variable 12 bp barcode sequences were used to
amplify the V4 region of the 16S rRNA gene (35). Primers 27F
(5'-AGRGTTYGATYMTGGCTCAG-3') and 1492R (5'-RGYTACCTTGTTACGACTT-3') were used
for the full-length (V1-V9) 16S rRNA gene amplification (LoopSeq and PacBio sequence) (36). The

qPCR reaction system included 22.5 μ L PCR SuperMix, 1.0 μ L positive primer and 1.0 μ L reverse
primer, 10 ng template DNA, dd H₂O supplement to 25 μ L. The amplification program was 1 cycle of
95 °C for 60 s, 28 cycles of 95 °C for 60 s, annealing at 58 °C for 60 s and primer extension at 72 °C
for 2 min, and finally 1 cycle of 72 °C for 10 min. Both V4 and full-length sequencing were performed
according to HUADA's standard procedures. Illumina Miseq platform was used to sequence the V4
amplicon (reagent kit v.3; Illumina). The full-length sequencing of PacBio was completed on the
Pacbio RS II platform.

Loopseq-16s-microorganism-24-plex-kit (Loop Genomics, San Jose, CA, USA) was used to
analyze the microbial genome of rhizosphere soil. The unique molecular markers of a single 16S gene
used in LoopSeq sequencing were distributed in the whole gene, and then the full-length 16S gene was
recombined through short reading and sequencing on the Illumina platform. Briefly, 10ng DNA from
different rhizosphere soil samples was used to build a sequencing library. The raw data were collected
by Illumina's NextSeq, with generated FASTQ file (37). All the raw data of the V4 16S sequence,

[revised manuscript text omitted]

548 soybean varieties.

549 Contribution

550 L.T., and N.H. conceived and designed the experiments. Y.T., L.T., C.L., L.Q., and W.S. performed the
551 experiments. Z.Y., Z.H., and T.M. completed the library preparation and sequencing. Y.T., L.T., and N.H.
analyzed the data and wrote the paper.

Declaration of conflicts

We declare no conflicts of interest.

References

- 1. Boyer JS, Byrne P, Cassmand KG, Cooper M, Delmer D, Greene T, Gruis F, Habben J,
Hausmann N, Kenny N, Lafitte R, Paszkiewicz S, Porter D, Schlegel A, Schussler J, Setter T,
Shanahan J, Sharp RE, Vyn TJ, Warner D, Gaffney J. 2013. The U.S. drought of 2012 in
perspective: A call to action. *Glob Food Secur-Agr* 2(3): 139–143.
<https://doi.org/10.1016/j.gfs.2013.08.002>.
- 2. Lesk C, Rowhani P, Ramankutty N. 2016. Influence of extreme weather disasters on global crop
production. *Nature* 529(7584): 84–87. <https://doi.org/10.1038/nature16467>.
- 3. Dennis ES, Dolferus R, Ellis M, Muhammad HR, Yongrui W, Hoeren FU, Anil G, Kathleen PI,
Allen GG, Peacock WJ. 2000. Molecular strategies for improving waterlogging tolerance in plants.
*J Exp Bot* 51: 79–82. <https://doi.org/10.1093/jexbot/51.342.89>.
- 4. Kreuzwieser J, Rennenberg H. 2014. Molecular and physiological responses of trees to
waterlogging stress. *Plant Cell Environ* 37(10): 2245–2259. <https://doi.org/10.1111/pce.12310>.
- 5. Sairam RK, Kumutha D, Ezhilmathi K, Chinnusamy V, Meena RC. 2009. Waterlogging induced
oxidative stress and antioxidant enzyme activities in pigeon pea. *Biol Plantarum* 53: 493–504.
<https://doi.org/10.1007/s10535-009-0090-3>.
- 6. Grichko VP, Glick BR. 2001. Amelioration of flooding stress by ACC deaminasecontaining plant
growth-promoting bacteria. *Plant Physiol Bioch.* 39(1): 11–17.
[https://doi.org/10.1016/S0981-9428\(00\)01212-2](https://doi.org/10.1016/S0981-9428(00)01212-2).
- 7. Yang H, Sheng R, Zhang Z, Ling W, Qing W, Wei WX. 2016. Responses of nitrifying and
denitrifying bacteria to flooding drying cycles in flooded rice soil. *Appl Soil Ecol* 103: 101–109.
<https://doi.org/10.1016/j.apsoil.2016.03.008>.
- 8. Neatrou MA, Webster JR, Benfield EF. 2004. The role of floods in particulate organic matter
dynamics of a southern Appalachian river-floodplain ecosystem. *J North Am Benthol Soc.* 23(2):
198–213. [https://doi.org/10.1899/0887-3593\(2004\)023<0198:TROFIP>2.0.CO;2](https://doi.org/10.1899/0887-3593(2004)023<0198:TROFIP>2.0.CO;2).
- 9. Sánchez-Rodríguez AR, Hill PW, Chadwick DR, Jones DL. 2019. Typology of extreme flood
event leads to differential impacts on soil functioning. *Soil Biol Biochem* 129: 153–168.
<https://doi.org/10.1016/j.soilbio.2018.11.019>.

- 10. Ngumbi E, Kloepper J. 2016. Bacterial-mediated drought tolerance: Current and future prospects.
*Appl Soil Ecol* 105: 109–125. <https://doi.org/10.1016/j.apsoil.2016.04.009>.
- 11. Kang SM, Khan AL, Waqas M, Young HY, Jin HK, Jong GK, Muhammad H, In JL. 2014. Plant
growth-promoting rhizobacteria reduce adverse effects of salinity and osmotic stress by regulating
phytohormones and antioxidants in *Cucumis sativus*. *J Plant Interact* 9: 673–682.
<https://doi.org/10.1080/17429145.2014.894587>.
- 12. Nascimento FX, Rossi MJ, Soares CR, Brendan JM, Bernard RG. 2014. New insights into
1-aminocyclopropane-1-carboxylate (ACC) deaminase phylogeny evolution and ecological
significance. *PLoS One* 9(6): e99168. <https://doi.org/10.1371/journal.pone.0099168>.
- 13. Armada E, Azcon R, Lopez C, Calvo PM, Ruiz-Lozano JM. 2015. Autochthonous arbuscular
mycorrhizal fungi and *Bacillus thuringiensis* from a degraded Mediterranean area can be used to
improve physiological traits and performance of a plant of agronomic interest under drought
conditions. *Plant Physiol Bioch.* 90: 64–74. <https://doi.org/10.1016/j.plaphy.2015.03.004>.
- 14. Lakshmanan V, Ray P, Craven KD. 2017. Toward a resilient functional microbiome: drought
tolerance-alleviating microbes for sustainable agriculture. *Methods Mol Biol* 1631: 69–84.
https://doi.org/10.1007/978-1-4939-7136-7_4.
- 15. Mentzer JL, Goodman RM, Balsler TC. 2006. Microbial response over time to hydrologic and
fertilization treatments in a simulated wet prairie. *Plant Soil* 284: 85–100.
<https://doi.org/10.1007/s11104-006-0032-1>.
- 16. Suzuki C, Kunito T, Aono T, Liu CT, Oyaizu H. 2005. Microbial indices of soil fertility. *J Appl*
*Microbiol* 98: 1062–1074. <https://doi.org/10.1111/j.1365-2672.2004.02529.x>.
- 17. Evans SE, Wallenstein MD. 2012. Soil microbial community response to drying and rewetting
stress: does historical precipitation regime matter. *Biogeochemistry* 109: 101–116.
<https://doi.org/10.2307/41490547>.
- 18. Kozich JJ, Westcott SL, Baxter NT, Sarah KH, Patrick DS. 2013. Development of a dual-index
sequencing strategy and curation pipeline for analyzing amplicon sequence data on the miseq
illumina sequencing platform. *Appl Environ Microb.* 79(17): 5112–5120.
<https://doi.org/10.1128/AEM.01043-13>.
- 19. Schloss PD, Eisen JA. 2010. The effects of alignment quality distance calculation method
sequence filtering and region on the analysis of 16S rRNA gene-based studies. *PLOS Comput*

- Biol 6(7): e1000844. <https://doi.org/10.1371/journal.pcbi.1000844>.
- 20. Youssef N, Sheik CS, Krumholz LR, Fares ZN, Bruce AR, Mostafa SE. 2009. Comparison of
species richness estimates obtained using nearly complete fragments and simulated
pyrosequencing-generated fragments in 16S rRNA gene-based environmental surveys. *Appl*
*Environ Microbiol* 75(16): 5227–5236. <https://doi.org/10.1128/AEM.00592-09>.
- 21. Gilbert JA, Meyer F, Antonopoulos D, Pavan B, Brown CT, Christopher TB, Narayan D, Jonathan
AE, Dir KE, Dawn F, Wu F, Daniel H, Janet J, Rob K, James K, Eugene K, Kostas K, Joel K,
Nikos K, Rachel M, Alice M, Christopher Q, Jeroen R, Alexander S, Ashley S, Rick S. 2010.
Meeting report: the terabase metagenomics workshop and the vision of an Earth microbiome
project. *Stand Genomic Sci.* 3(3): 243–248. <https://doi.org/10.4056/sigs.1433550>.
- 22. Huse SM, Dethlefsen L, Huber JA, Welch DM, Relman DA, Sogin ML. 2008. Exploring
microbial diversity and taxonomy using SSU rRNA hypervariable tag sequencing. *PLoS Genet.*
4(11): e1000255. <https://doi.org/10.1371/journal.pgen.1000255>.
- 23. Claesson MJ, O'Sullivan O, Wang Q, Janne N, Julian RM, Hauke S, Willem MV, Ross RP,
O'Toole PW. 2009. Comparative analysis of pyrosequencing and a phylogenetic microarray for
exploring microbial community structures in the human distal intestine. *PLoS One* 4(8): e6669.
<https://doi.org/10.1371/journal.pone.0006669>.
- 24. Esther S, Brian B, Devin CD, Brett B, Robert MB, Asaf L, Esther AG, Jan FC, Alex C, Hans PK,
Steven JH, Philip H, Susannah GT, Tanja W. 2016. High-resolution phylogenetic microbial
community profiling. *ISME J.* 10: 2020–2032. <https://doi.org/10.1038/ismej.2015.249>.
- 25. Tremblay J, Singh K, Fern A, Kirton ES, Tringe SG, Woyke T, Janey L, Feng C, Jeffery LD,
Susannah GT. 2015. Primer and platform effects on 16S rRNA tag sequencing. *Front Microbiol* 6:
771. <https://doi.org/10.3389/fmicb.2015.00771>.
- 26. Guo F, Ju F, Cai L, Zhang T. 2013. Taxonomic precision of different hypervariable regions of 16S
rRNA gene and annotation methods for functional bacterial groups in biological wastewater
treatment. *PLoS One* 8(10): e76185. <https://doi.org/10.1371/journal.pone.0076185>.
- 27. Earl JP, Adappa ND, Krol J, Bhat AS, Balashov S, Ehrlich RL, Palmer JN, Workman AD, Blasetti
640 M, Sen B, Hammond J, Cohen NA, Ehrlich GD, Mel JC. 2018. Species-level bacterial community
profiling of the healthy sinonasal microbiome using Pacific Biosciences sequencing of full-length
16S rRNA genes. *Microbiome* 6(1): 190. <https://doi.org/10.1186/s40168-018-0569-2>.

[revised manuscript text omitted]

81. Hirokazu T, Kabir GP, Masato Y, Kazuhiko N, Kei H, Ken N, Shinji F, Masayuki U, Shinji N,
Yusuke O, Kentaro Y, Klaus S, Yang B, Ryo S, Yasunori I, Kiwamu M, Kiers ET. 2018. Core
microbiomes for sustainable agroecosystems. *Nat Plants* 4: 247–257.
<https://doi.org/10.1038/s41477-018-0139-4>.

82. Xun W, Liu Y, Li W, Yi R, Wu X, Xu ZH, Nan Z, Miao YZ, Shen QR, Zhang RF. 2021.
Specialized metabolic functions of keystone taxa sustain soil microbiome stability. *Microbiome*
9(1): 35. <https://doi.org/10.1186/s40168-020-00985-9>.

83. Anne D, Katharina K, Hanna K, Craig WH, Michaela S, Jasmin S, Thomas Z, Søren MK, Mads A,
Per HN, Michael W, Holger D. 2020. Exploring the upper pH limits of nitrite oxidation: diversity
ecophysiology and adaptive traits of haloalkaliphilic Nitrospira. *ISME J* 14: 2967–2979.
<https://doi.org/10.1038/s41396-020-0724-1>.

84. Zhong YQW, Hu JH, Xia QM, Zhang SL, Xin L, Pan XY, Zhao RP, Wang RW, Yan WM, Shang
GZP, Hu FY, Yang CD, Wen W. 2020. Soil microbial mechanisms promoting ultrahigh rice yield.
*Soil Biol Biochem* 143: 107741. <https://doi.org/10.1016/j.soilbio.2020.107741>.

85. Angana S, Sufia KK, Pinaki S. 2014. Studies on arsenic transforming groundwater bacteria and
their role in arsenic release from subsurface sediment. *Environ. Sci Pollut R* 21: 8645–8662.

- <https://doi.org/10.1007/s11356-014-2759-1>.
- 86. Zhang JY, Liu YX, Guo XX, Qin Y, Garrido-Oter R, Schulze-Lefert P, Bai Y. 2021.
High-throughput cultivation and identification of bacteria from the plant root microbiota. *Nat*
*Protocols* 16: 988–1012. <https://doi.org/10.1038/s41596-020-00444-7>.
- 87. Vorholt JA, Vogel C, Carlström CI, Müller DB. 2017. Establishing causality: opportunities of
synthetic communities for plant microbiome research. *Cell Host Microbe* 22(2): 142–155.
<https://doi.org/10.1016/j.chom.2017.07.004>.
- 88. Faust K, Raes J. 2012. Microbial interactions: from networks to models. *Nat Rev Microbiol* 10:
538–550. <https://doi.org/10.1038/nrmicro2832>.
- 89. Ziegler M, Eguíluz VM, Duarte CM, Voolstra CR. 2018. Rare symbionts may contribute to the
resilience of coral–algal assemblages. *ISME J* 12(1): 161–72.
<https://doi.org/10.1038/ismej.2017.151>.
- 90. Kara EL, Hanson PC, Hu YH, Luke W, Katherine DM. 2013. A decade of seasonal dynamics and
co-occurrences within freshwater bacterioplankton communities from eutrophic Lake Mendota
WI USA. *ISME J* 7(3): 680–684. <https://doi.org/10.1038/ismej.2012.118>.
- 91. Olesen JM, Bascompte J, Dupont YL, Jordano P. 2007. The modularity of pollination networks.
*PNAS* 104: 19891–19896. <https://doi.org/10.1073/pnas.0706375104>.
- 92. Wu X, Alexandre J, Sai G, Ida K, Zhao QY, Wu HS, George AK, Shen QR, Li R, Stefan G. 2018.
Soil protist communities form a dynamic hub in the soil microbiome. *ISME J* 12: 634–638.
<https://doi.org/10.1038/ismej.2017.171>.
- 93. Chen S, Tatoba RW, Ruibo S, Kuramae EE, Liu B. 2019. Root-associated microbiomes of wheat
under the combined effect of plant development and nitrogen fertilization. *Microbiome* 7(1): 136.
<https://doi.org/10.1186/s40168-019-0750-2>.
- 94. Lalucat J, Bennasar A, Bosch R, Elena GV, Norberto JP. 2006. Biology of *Pseudomonas stutzeri*.
*Microbiol. Mol Biol Rev* 70(2): 510–547. <https://doi.org/10.1128/MMBR.00047-05>.
- 95. Patricia CDS, Dennis RD. 2011. Co-ordination and fine-tuning of nitrogen fixation in *Azotobacter*
*vinelandii*. *Mol Microbiol* 79: 1132–1135. <https://doi.org/10.1111/j.13>.
- 96. Lisa R, Karoline F. 2018. From hairballs to hypotheses-biological insights from microbial
networks. *Fems Microbiol Rev* 42(6): 761–780. <https://doi.org/10.1093/femsre/fuy030>.
- 97. Hirano H, Takemoto K. 2019. Difficulty in inferring microbial community structure based on

co-occurrence network approaches. *BMC Bioinformatics* 20(1): 329.
<https://doi.org/10.1186/s12859-019-2915-1>.

98. Baskaran V, Patil PK, Antony ML, Avunje S, Nagaraju VT, Ghate SD, Nathamuni S,
Dineshkumar N, Alavandi SV, Vijayan KK. 2020. Microbial community profiling of ammonia and
nitrite oxidizing bacterial enrichments from brackishwater ecosystems for mitigating nitrogen
species. *Sci Rep-UK* 10: 5201. <https://doi.org/10.1038/s41598-020-62183-9>.

99. Liao SL, Wang YY, Liu H, Fan GY, Sunil KS, Tao J, Chen JW, Zhang PF, Lone G, Mikael LS, Shi
Q, Simon M, Yuen L, Liu X. 2020. Deciphering the microbial taxonomy and functionality of two
diverse mangrove ecosystems and their potential abilities to produce bioactive compounds.
*mSystems* 5(5): e00851–19. <https://doi.org/10.1128/mSystems.00851-19>.

100. Xu J, Liu SJ, Song SR, Guo HL, Tang JJ, Yong JWH, Ma YD, Chen X. 2019. Arbuscular
mycorrhizal fungi influence decomposition and the associated soil microbial community under
different soil phosphorus availability. *Soil Biol. Biochem.* 120: 181–190.
<https://doi.org/10.1016/j.soilbio.2018.02.010>.

101. Ma B, Wang H, Dsouza M, Jun L, Yan H, Dai ZM, Philip CB, Xu JM, Jack AG. 2015. Geographic
patterns of co-occurrence network topological features for soil microbiota at continental scale in
eastern China. *ISME J* 10: 1891–1901. <https://doi.org/10.1038/ismej.2015.261>.

102. Fiona MS, Paul BLG, Inma L, Davey LJ, Simon C, David AR. 2020. Soil textural heterogeneity
impacts bacterial but not fungal diversity. *Soil Biol Biochem* 144: 107766.
<https://doi.org/10.1016/j.soilbio.2020.107766>.

103. Trivedi P, Leach JE, Tringe SG, Sa T, Singh BK. 2021. Plant - microbiome interactions: from
community assembly to plant health. *Nat Rev Microbiol* 18(11): 607–621.
<https://doi.org/10.1038/s41579-020-0412-1>.

104. Tao K, Kelly S, Radutoiu S. 2019. Microbial associations enabling nitrogen acquisition in plants.
*Curr Opin Microbiol* 49: 83–89. <https://doi.org/10.1016/j.mib.2019.10.005>.

105. Yu XJ. Chen Q. Shi WC. Gao Z. Sun X. Dong JJ. Li J. Wang HT. Gao JG. Liu ZG. Zhang M.
2021. Interactions between phosphorus availability and microbes in a wheat-maize double
cropping system: a reduced fertilization scheme. *J Integr Agr* 20(0): 2–16.
[https://doi.org/10.1016/S2095-3119\(20\)63599-7](https://doi.org/10.1016/S2095-3119(20)63599-7).

106. Meyer CA, Liu XS. 2014. Identifying and mitigating bias in next-generation sequencing methods

for chromatin biology. *Nat Rev Genet* 15(11): 709–721. <https://doi.org/10.1038/nrg3788>.

107. Schirmer M, Ijaz UZ, D'Amore R, Hall N, Sloan WT, Quince C. 2015. Insight into biases and

sequencing errors for amplicon sequencing with the Illumina MiSeq platform. *Nucleic Acids Res.*

43 (6): e37. <https://doi.org/10.1093/nar/gku1341>.

Table 1 Related attributes of different sequencing platforms

	V4	LoopSeq	PacBio
Cloning required	No	No	No
Average sequence time	8h	2.5h	2 h/SMRT cell
Average read length	~250bp	~1500bp	1000-1500bp
Read technology	Short-read technology	Long-reads technology	Long-reads technology
Sequencing variable region	V4	V1-V9	V1-V9
Stitching during sequencing	Yes	No	No
Error rates	high	lower	medium
Eliminate PCR bias	NO	Yes	Yes
Need stitching	Yes	NO	NO
Approximate cost per Mb	US\$0.11	US\$0.245	US\$2.50

Table 2 Effects of waterlogging and soil types on bacterial community structure in soybean
 rhizosphere analyzed by permutational multivariate analysis of variance (PERMANOVA).

Sequencing methods	Factor	F	R ²	P
V ₄	Soil	111.7140	0.66882	0.001 ***
	Water	10.2152	0.06116	0.002 **
	Soil:Water	1.1018	0.00660	0.246
	NeW Vs NeCK	7.0589	0.24292	0.001 ***
	AcW Vs AcCK	6.7174	0.23391	0.001 ***
LooSeq	Soil	17.5165	0.26413	0.001 ***
	Water	2.6542	0.04002	0.012 *
	Soil:Water	2.1465	0.03237	0.026 *
	NeW Vs NeCK	2.5809	0.105	0.002 **
	AcW Vs AcCK	3.0537	0.12189	0.001 ***
PacBio	Soil	0.9707	0.01801	0.358
	Water	7.9983	0.14838	0.001 ***
	Soil:Water	0.9332	0.01731	0.386
	NeW Vs NeCK	2.5963	0.10556	0.001 ***
	AcW Vs AcCK	7.9223	0.26476	0.001 ***

* indicates significant value of $P < 0.05$, ** indicates significant value of $P < 0.01$, *** indicates
 significant value of $P < 0.001$.

Fig. 1. Effects of waterlogging and soil types on soybean rhizosphere soil bacterial (A) Chao1 and (B)
Shannon index with Illumina Miseq, LoopSeq, and PacBio full-length sequencing method (Dunn's t
tests, $n=6$, $P < 0.05$). (CDE) Constrained Principal-coordinate analysis (CPCoA) based on Bray-Curtis
distance showing differences in rhizosphere bacterial community structure under waterlogging in
neutral and acidic soil (PERMANOVA, $n=6$, $P < 0.05$). Different letters indicate significant
differences ($P < 0.05$). Ne: neutral soil; Ac: acidic soil; W: waterlogging; CK: without waterlogging;
NeCK: Soybean rhizosphere soil without waterlogging in neutral soil; NeW: Soybean rhizosphere soil
with waterlogging in neutral soil; AcCK: Soybean rhizosphere soil without waterlogging in acidic soil;
AcW: Soybean rhizosphere soil with waterlogging in acid soil. V4: Illumina Miseq; LoopSeq:
Full-length Loop Genomics sequencing technology; PacBio: Full-length the PacBio single molecule,
real-time (SMRT) technology.

Fig. 2. Taxonomy profiles in different sequencing methods datasets. The proportion of annotation
sequences from the V4 ($n=48$, blue), LoopSeq ($n=48$, yellow), and PacBio ($n=48$, orange) datasets
was determined by comparing the sequence with the SILVA database, and are represented at the (A)
phylum, (C) genus, and (E) species levels. Venn diagram showing the numbers of unique and shared
(B) phylum, (D) genus, and (F) species between three sequencing methods. Blue denotes V4, yellow
denotes LoopSeq, and orange denotes PacBio.

Fig. 3. Relative abundance analysis of common genus in three sequencing methods. The most
abundant 30 genera were selected from shared 201 genera of the three sequencing methods. Color
pairs denote samples of three sequencing methods in neutral or acidic soil with different waterlogging
917 times. Bubble sizes indicate the averaged relative abundance of an individual genus across each
918 treatment. The explanation of the abbreviations of different treatments were the same as in the legend
to Figure 1.

Fig. 4. Rhizosphere core microorganisms of different sequencing methods. The different parts inside

the double pie chart represent the bacterial phyla of the soybean core microbiome. The different parts
outside the double pie chart represent the OTU (genus) of the soybean core microbiome, and each
OTU (genus) is assigned to the corresponding bacterial phyla. The size of the different double pie
chart portions represents the percentage of phylum/genus relative abundance in all core microbial
components.

Fig. 5. (A, B, C) Network analysis reveals the symbiotic pattern between OTUs. The nodes are colored
according to the modular type. The connections between nodes indicate strong and significant
(Spearman's $r > 0.8$ or $r < -0.8$) ($P < 0.01$) correlation. The volcano map shows the amount of OTU
enriched and depleted in neutral soil and after waterlogging in the modules of different sequencing
methods, respectively. Violet denote Module I, green denotes Module II, blue denotes Module III,
black denotes Module IV, orange denotes Module V, red denotes Module VI, cyan denotes Module VII,
grey denotes other modules.

Fig. 6. Paired comparison of environmental factors and microbial community with a color gradient
denoting Pearson's correlation coefficient. Spearman's correlation coefficient > 0 indicates positive
correlation and < 0 indicates negative correlation. Effects of environmental factors in (A) two types of
soil, (B) neutral soil, (C) and acidic soil on the microbial communities of the three sequencing
methods. The edge width corresponds to the distance dependence of Mantel's R statistic, and
Statistical significance based on 9,999 permutations represents edge color. Mantel's r size indicates the
strength of the correlation. The color of the connecting line indicates the correlation between different
sequencing methods and environmental factors.

B

LoopSeq

C

PacBio

A

V4

- Module I (35.88%)
- Module II (32.76%)
- Module III (28.45%)
- Module IV (0.83%)
- Module V (0.35%)
- Module VI (0.21%)
- Module VII (0.21%)
- Other modules (1.31%)

B

LoopSeq

- Module I (32.98%)
- Module II (14.61%)
- Module III (5.42%)
- Module IV (4.67%)
- Module V (1.96%)
- Module VI (1.81%)
- Module VII (1.81%)
- Other modules (36.82%)

C

PacBio

- Module I (28.82%)
- Module II (22.94%)
- Module III (20.59%)
- Module IV (20%)
- Module V (1.18%)
- Module VI (1.18%)
- Module VII (1.18%)
- Other modules (4.11%)